# Transformers can optimally learn regression mixture models

**Reese Pathak**\*
Department of Electrical Engineering and Computer Sciences
University of California, Berkeley
Berkeley, CA 94709, USA
`pathakr@eecs.berkeley.edu`

**Rajat Sen, Weihao Kong, & Abhimanyu Das**
Google Research
Mountain View, CA 94043, USA

## Abstract

Mixture models arise in many regression problems, but most methods have seen limited adoption partly due to these algorithms' highly-tailored and model-specific nature. On the other hand, transformers are flexible, neural sequence models that present the intriguing possibility of providing general-purpose prediction methods, even in this mixture setting. In this work, we investigate the hypothesis that transformers can learn an optimal predictor for mixtures of regressions. We construct a generative process for a mixture of linear regressions for which the decision-theoretic optimal procedure is given by data-driven exponential weights on a finite set of parameters. We observe that transformers achieve low mean-squared error on data generated via this process. By probing the transformer's output at inference time, we also show that transformers typically make predictions that are close to the optimal predictor. Our experiments also demonstrate that transformers can learn mixtures of regressions in a sample-efficient fashion and are somewhat robust to distribution shifts. We complement our experimental observations by proving constructively that the decision-theoretic optimal procedure is indeed implementable by a transformer.

## 1 Introduction

In several machine learning applications—federated learning (Wang et al., 2021), crowd-sourcing (Steinhardt et al., 2016) and recommendations systems (Wang et al., 2006)—data is collected from multiple sources. Each source generally provides a small batch of data: for instance in recommendation systems, a user can provide a source of rating data on a subset of items that she has encountered. Such batches, on their own, are often too small to learn an effective model for the desired application. On the other hand, by pooling many batches together, improvements can typically be made in the quality of the predictors that can be learned.

An issue with this "pooling" approach is that if it is done carelessly, then the models which are learned may lack personalization (Ting et al., 1999). For instance, in a recommendation system, such an approach could yield a model that selects similar actions for dissimilar users. A better approach, however, is to model the problem as a *mixture* of distributions: for instance, we can model the sources as arising from $m$ subpopulations, assuming that sources arising from the subpopulation have similar underlying distributions (Kleinberg & Sandler, 2004). The sources from a single subpopulation can then be aggregated for the purposes of learning. For instance, in the recommendation systems example, users in the same subpopulation might be identified as having similar preferences and tastes for item genres.

---

\*Work was completed during an internship at Google Research

A supervised learning formulation of the above setup is that the sources arise from a subpopulation indexed by an integer $i \in [m] := \{1, 2, \cdots, m\}$. Additionally, assume that within each subpopulation the input-output pair $(x, y)$ follows a relation of the form $y = f_i^\star(x) + \eta$, where $\eta$ is a zero-mean noise, and $x \in \mathbf{R}^d$. A batch of i.i.d data from such a source can be represented as $\{(x_j, y_j)\}_{j=1}^k$ where $k$ is the batch size. Given many such batches, each having examples only from one source, the task is to learn the functions $\{f_i^\star\}_{i=1}^m$ well enough to make good predictions on another input, sometimes also referred to as a *query*, $x_{k+1}$. For instance, given the past ratings of an user, we should be able to determine their subpopulation well enough to infer their preferences on an unseen item.

The simplest version of the formulation above additionally imposes the assumption that the functions $f_i^\star$ are linear: $f_i^\star(x) = \langle w_i^\star, x_i \rangle$. This setting has been studied theoretically in (Kong et al., 2020; Jain et al., 2023). Kong et al. (2020) introduced the problem and designed an algorithm in the setting where there are as many as $O(d)$ batches with size $k = O(1)$, and fewer medium-sized batched of size $k = O(\sqrt{m})$. However, that work imposed strong assumptions on the covariate distribution, which lead to the paper (Jain et al., 2023), where these assumptions were relaxed. This latter work proposes a different algorithm that even allows covariate distributions to vary among subpopulations. Nonetheless, their algorithm needs to know problem parameters, such as a $L^2 - L^4$ hypercontractivity, a spectral norm bound on the covariance matrix, the noise level, and the number of subpopulations $m$. It is not clear how these algorithms will fare with model misspecification, or if they could be extended to applications like federated learning where it is unlikely that the correct model is linear, and distributed algorithms are required.

In this work, we ask the question: *Is there a deep learning architecture that can be trained using standard gradient decent, yet learns mixture models from batched data and can leverage small batches from a source to make predictions for its appropriate subpopulation?* If so, this would allow us to solve these type of mixture problem without needing highly specialized algorithms that could potentially be brittle with respect to knowing the correct form of the mixture model. Moreover, standard (stochastic) gradient descent would naturally extend to distributed training, using standard techniques from federated learning (Wang et al., 2006).

A natural candidate to address this question is the widely used transformer architecture (Vaswani et al., 2017). Motivated by their immense success in NLP (Radford et al., 2019), computer vision (Han et al., 2022) and in context learning abilities demonstrated by large models (Chowdhery et al., 2022), several recent works have been aimed to analyze whether transformers can learn algorithms (Akyürek et al., 2023; Garg et al., 2022; Von Oswald et al., 2023). These papers train decoder only transformers using prompts of the form $(x_1, f(x_1), \cdots, x_j, f(x_j), \cdots x_k, f(x_k), x_{k+1})$ where the task is to predict $f(x_{k+1})$ after seeing the portion of the prompt preceding it. These papers show empirically that when $f$ is sampled from a class of linear functions, then transformers learn to perform linear regression in-context. Akyürek et al. (2023) also show that transformers can represent gradient descent in the case of linear regression provided through a construction.

Another important motivation for studying mixture models in the context of transformers is the question of data mixtures for pretraining large language models (LLM's). It is well known that for LLM's to succeed in various in-context learning tasks the pretraining dataset needs to be diverse, covering various languages, tasks, programming languages and concepts. This is evident from the complex datasheets for training popular LLMs (Chowdhery et al., 2022). A natural question then is to investigate whether transformers can zero in on relevant parts of the pretraining data given a new prompt and model select between algorithms (Bai et al., 2023). The mixture model studied in this paper can be seen as a simple linear or nonlinear analogue of the smae model.

**Our contributions:**

- We demonstrate that transformers can learn mixtures of linear regressions by training on such mixture data and exhibiting near-Bayes-optimal error at inference time.

- We strengthen this observation by proving constructively that transformers can implement the optimal method for the mixture distribution on which the transformer was trained.

- Our experiments show that transformers are sample-efficient: the transformers' performance is similar (or better) than model-specific methods, when fixing the training set size.

- We evaluate certain inference-time metrics that capture the nearness of predictions made by the transformer versus another predictor. We show these metrics are smallest when taking the comparator to be the decision-theoretic optimal method, thereby further corroborating the hypothesis that transformers optimally learn mixtures of regressions.
- We also show that the above empirical observations carry over to settings where $f_i^*$'s are non-linear specifically when they are polynomials or multi-layer perceptron (MLP)'s with two hidden layers.
- We suggest that transformers tolerate "small" distribution shifts by investigating transformers' performance on both covariate and label shifts to the mixture model.

These contributions, taken together, are evidence that transformers can optimally, efficiently, and robustly learn mixtures of linear regressions.

**Related work:** The related work can be broadly divided into a thread that studies the theoretical properties of algorithms for estimation or prediction in a regression mixture model as well as another thread that studies the empirical and theoretical properties of transformers on learning regression models. Due to space considerations, we present a more detailed overview in Appendix A.

## 1.1 GENERATIVE MODEL FOR DATA

Throughout this paper we consider mixture of linear regression, except in Appendix E, where an extension to nonlinear models is considered. Underlying the mixture of linear regressions, we consider the discrete mixture

$$\pi := \frac{1}{m} \sum_{i=1}^{m} \delta_{w_i^\star}, \tag{1}$$

where $\{w_i^\star\}_{i=1}^m \in \mathbf{R}^d$ are normalized such that $\|w_i^\star\|_2 = \sqrt{d}$ for each $i \in [m]$. We consider *prompts* or *batches*, denoted $P = (x_1, y_1, \ldots, x_k, y_k, x_{k+1})$. Here, for noise level $\sigma \geqslant 0$, we have

$$w \sim \pi, \quad x_i \overset{\text{i.i.d.}}{\sim} \mathsf{N}(0, I_d), \quad \text{and} \quad y_i \mid x_i \sim \mathsf{N}(\langle w, x_i \rangle, \sigma^2).^1 \tag{2}$$

The goal is then to predict $y_{k+1}$, the label for the query $x_{k+1}$.

## 1.2 TRANSFORMERS

Transformers are deep neural networks that map sequences to sequences (Vaswani et al., 2017). In this work, we specifically focus on decoder-only, *autoregressive transformers*. These models are comprised of multiple layers that map an input matrix $H \in \mathbf{R}^{p \times q}$ to an output matrix $H' \in \mathbf{R}^{p \times q}$. Here $p$ denotes the hidden dimension, and $q$ is corresponds to the number of input tokens. The output is then fed successively to more such layers. Since the computation in each layer is the same (apart from parameters), we describe the computation occurring in a single layer. Write $h_j = (H_{ij})_{1 \leqslant i \leqslant p}$ for the $j$th column of $H$, and $h_j'$ for the $j$th column of $H'$. Additionally, the prefix matrix $H_{:i}$ is the $p \times (i-1)$ submatrix of $H$ obtained by concatenating the first $i-1$ columns of $H$.[2]

A layer is parameterized by a sequence of weights. Let $n_{\text{heads}}$ denote the number of attention heads and $d_{\text{att}}$ denote the hidden dimension for the attention layer and $d_{\text{ff}}$ denote the hidden dimension for the feedforward (*i.e.*, dense) layer. Then, a layer has the following weights:

$$\{W_i^{\mathrm{Q}}\}_{i=1}^{n_{\text{heads}}}, \{W_i^{\mathrm{V}}\}_{i=1}^{n_{\text{heads}}}, \{W_i^{\mathrm{K}}\}_{i=1}^{n_{\text{heads}}}, \subset \mathbf{R}^{d_{\text{att}} \times p},$$
$$\{W_i^{\mathrm{C}}\}_{i=1}^{n_{\text{heads}}} \subset \mathbf{R}^{p \times d_{\text{att}}} \quad W^{\mathrm{in}} \in \mathbf{R}^{d_{\text{ff}} \times p}, \quad \text{and} \quad W^{\mathrm{out}} \in \mathbf{R}^{p \times d_{\text{ff}}} \tag{3}$$

for each column $i \in [p]$, the computation proceeds in the following fashion.

---

[1] When $\sigma = 0$, by $\mathsf{N}(v, 0)$ we mean the point mass $\delta_v$.

[2] In the case $i = 1$, the submatrix can be interpreted as 0.

**Self-attention:** The layer begins with computing the attention vector, $a_i \in \mathbf{R}^p$, by

$$s_{ij} := \text{softmax}\left(\left(W_j^{\text{K}} H_{:i}\right)^{\mathsf{T}} W_j^{\text{Q}} h_i\right), \quad \text{for } j \in [n_{\text{heads}}], \quad \text{and,}$$

$$a_i := \sum_{j=1}^{n_{\text{heads}}} W_j^{\text{C}} W_j^{\text{V}} H_{:i} s_{ij}$$

Above, with a slight abuse of notation, we define for any integer $\ell > 0$, softmax: $\mathbf{R}^\ell \to \mathbf{R}^\ell$ by the formula $\text{softmax}(v) = (e^{v_t} / \sum_{t'=1}^{\ell} e^{v_{t'}})_{t=1}^{\ell}$. Note that, above, $s_{ij} \in \mathbf{R}_+^{i-1}$.[3]

**Feedforward network:** The layer then continues by passing the attention vector (along with the original input column $h_i$) through a nonlinear dense layer. This is defined by

$$h_i' := a_i + h_i + W^{\text{out}} \sigma_{\bullet}(W^{\text{in}} \lambda(a_i + h_i))$$

Above the notation $\sigma_{\bullet}$ indicates that the map $\sigma \colon \mathbf{R} \to \mathbf{R}$ is applied *componentwise* to its argument. In this work we take the nonlinearity to be the Gaussian error linear unit (GeLU) (Hendrycks & Gimpel, 2016) which is defined by

$$\sigma(u) = \frac{u}{2}\left(1 + \text{erf}\left(\frac{u}{\sqrt{2}}\right)\right), \quad \text{for any } u \in \mathbf{R}.$$

Above, erf denotes the Gauss error function. The function $\lambda \colon \mathbf{R}^p \to \mathbf{R}^p$ denotes layer normalization (Ba et al., 2016), and is given by

$$\lambda(v) = \sqrt{p}\frac{v - \overline{v}\mathbf{1}_p}{\|v - \overline{v}\mathbf{1}_p\|_2}, \quad \text{where} \quad \overline{v} = \frac{1}{p}\sum_{i=1}^{p} v_i.$$

This is a form of standardization where $\overline{v}$ is interpreted as the mean (averaging the components) and $\|v - \overline{v}\mathbf{1}_p\|_2^2 / p$ is interpreted as the variance (averaging the squared deviation to the mean).

## 2 REPRESENTATION

In this section, we prove that transformers can actually represent the minimum mean squared error procedure. Indeed, let $f \colon P \mapsto \hat{y} \in \mathbf{R}$, by any procedure which takes a prompt $P$ and outputs an estimate $\hat{y}$ on the query, and define the mean squared error (MSE) by

$$\text{MSE}(f) := \mathbf{E}_P\left[(f(P) - y_{k+1})^2\right].$$

Then by standard Bayesian decision theory, under the observational model described in Section 1.1, it follows that the mean squared error is minimized at the posterior mean $f_{\pi}^{\star}$, which is given by

$$f_{\pi}^{\star}(P) = \langle \hat{w}(P), x_{k+1} \rangle \quad \text{where} \quad \hat{w}(P) := \frac{\sum_{j=1}^{m} w_j^{\star} \exp\left(-\frac{1}{2\sigma^2}\sum_{i=1}^{k}(\langle w_j^{\star}, x_i \rangle - y_i)^2\right)}{\sum_{\ell=1}^{m} \exp\left(-\frac{1}{2\sigma^2}\sum_{i=1}^{k}(\langle w_\ell^{\star}, x_i \rangle - y_i)^2\right)}. \quad (4)$$

Formally, $\text{MSE}(f) \geqslant \text{MSE}(f_{\pi}^{\star})$, for all (measurable) $f$. Note above that $\hat{w}$ does not depend on $x_{k+1}$.

Then our main result is that the function $f_{\pi}^{\star}$ can be computed by a transformer.

**Theorem 1.** *There is an autoregressive transformer which implements the function $f_{\pi}^{\star}$ in (4).*

See Section B for a proof of this claim.

For an illustration of the underlying idea behind Theorem 1, see Figure 1 for an arithmetic circuit that computes the function $f_{\pi}^{\star}$, in the case $m = 3, k = 2$. The objects $r_{ij}$ are residuals, defined as

$$r_{ij} = \langle w_j^{\star}, x_i \rangle - y_i \mathbf{1}\{i \neq k+1\}, \quad \text{for } i \in [k], \ j \in [m]. \quad (5)$$

The first layer computes the values $\{r_{ij}\}$, the second layer computes the squares of these values, the third layer computes the (scaled) sum of these values over the index $i$, which runs over the samples in the prompt, excluding the query. The fourth layer, computes the softmax of these sums,

---

[3]When $i = 1$, $s_{ij} = 0$.

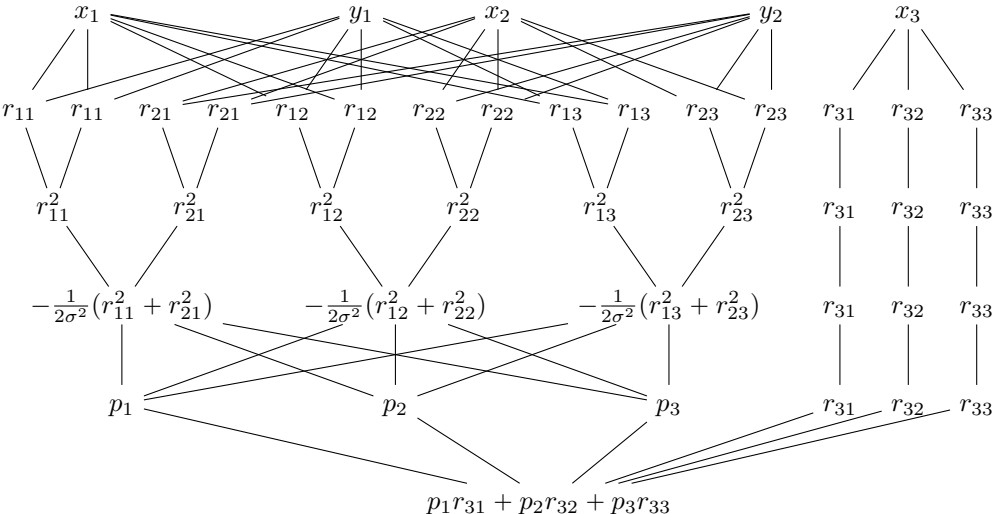

**Figure 1.** Illustration of an arithmetic circuit, implementable by a transformer, that computes the posterior mean as defined in display (4). Here, $r_{ij}$ are residuals as defined in display (5) and $p_j$ are probabilities obtained via a softmax operation, as defined in display (6). See main text for a description of the computation occurring at each level.

$$p_j := \frac{\exp\left(-\frac{1}{2\sigma^2}\sum_{i=1}^k (\langle w_j^\star, x_i\rangle - y_i)^2\right)}{\sum_{\ell=1}^m \exp\left(-\frac{1}{2\sigma^2}\sum_{i=1}^k (\langle w_\ell^\star, x_i\rangle - y_i)^2\right)}, \quad \text{for } j \in [m] \tag{6}$$

And the final layer computes

$$\sum_{j=1}^m p_j r_{(k+1),j} = \left\langle \sum_{j=1}^m p_j w_j^\star, x_{k+1}\right\rangle = \langle \hat{w}(P), x_{k+1}\rangle = f_\pi^\star(P)$$

where the last equation follows from the definitions in display (4). Therefore, the circuit depicted in Figure 1 is able to compute the posterior mean $f_\pi^\star$, at least for the choices $k = 2, m = 3$. Generalizing the circuit to general $(k, m)$ is straightforward; therefore, our proof amounts to exploiting the circuit and demonstrating that each operation: linear transforms in the first and final layers, squaring in the second layer, summation in the third layer, softmax in the fourth layers are all implementable by a transformer.

## 3 EXPERIMENTAL RESULTS

In this section, we present results of training transformers on batches as described in Section 1.1. Our methodology closely follows the training procedure described in (Garg et al., 2022). In the notation of Section 1.2, our transformer models set the hidden dimension as $p = 256$, feedforward network dimension as $d_{\text{ff}} = 4p = 1024$, and the number of attention heads as $n_{\text{heads}} = 8$. Our models have 12 layers. Additional details on the training methodology can be found in Appendix C. We also release our training and simulation code along with this paper.

### 3.1 TRANSFORMERS CAN LEARN MIXTURES OF LINEAR REGRESSIONS

To begin with, we investigate the performance of transformers on mixture models with various numbers of components and varying noise levels. We plot the performance of the transformer when prompted with a prompt $P$ of length $k$, for $1 \leqslant k \leqslant 60$. The normalized MSE is the mean-squared error between the true labels and the estimated labels, divided by the dimension $d = 20$. Above, the

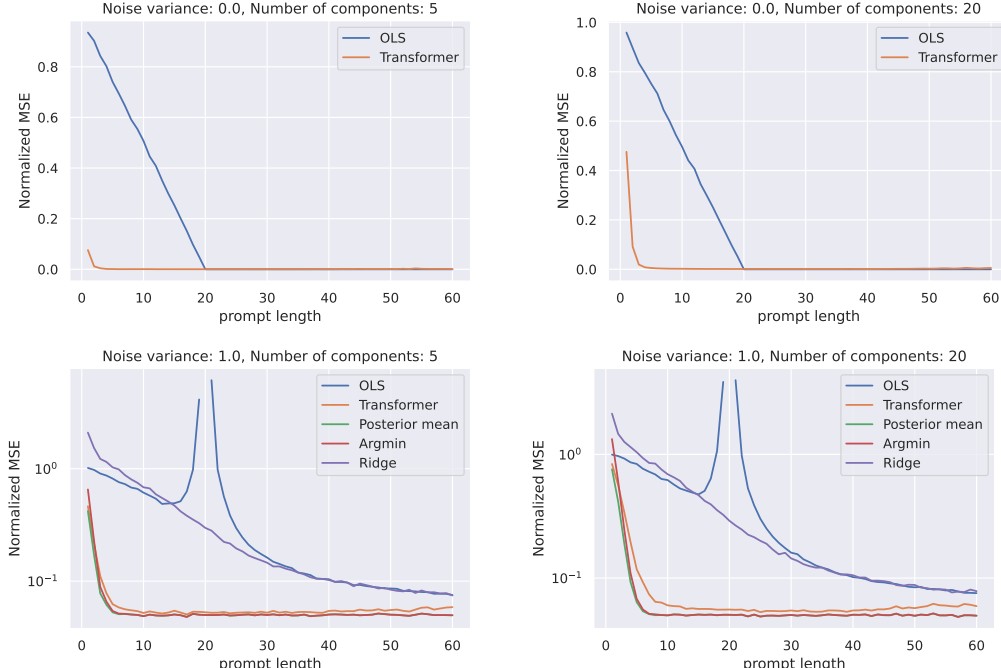

**Figure 2.** Transformer model trained on mixture of linear regressions data with 5 and 20 components. Top row is mixture data with no noise added to $y$ ($\sigma = 0$) and bottom row is mixture data with noise added ($\sigma = 1$). We also compare to ridge regression with the optimal penalty taken from either Corollary 3 in Dicker (2016) or Section 3.1.1 in Pathak et al. (2023).

algorithms that we compare against are:[4]

- *Ordinary least squares (OLS).* For a prompt of length $k$, computes an element $\hat{w} \in \arg\min_{w \in \mathbf{R}^d} \sum_{j \leqslant k} (w^\intercal x_j - y_j)^2$. Outputs $\hat{y}_{k+1} = \hat{w}^\intercal x_{k+1}$.

- *Posterior mean.* This is an oracle algorithm. Given a prompt $P$ of length $k$, computes the posterior mean $\hat{y}_{k+1} = f^\star_\pi(P)$, as defined in display (4).

- *Argmin.* This is an oracle algorithm. Given a prompt $P$ of length $k$, computes

$$\hat{w} = \arg\min_{w \in \{w^\star_j\}_{j=1}^m} \sum_{j \leqslant k} \sum_{j \leqslant k} (w^\intercal x_j - y_j)^2. \tag{7}$$

The prediction is then $\hat{y}_{k+1} = \hat{w}^\intercal x_{k+1}$

Strikingly, we see that the transformer predictions are as good as—or nearly as good as—the oracle procedures which have knowledge of the true mixture components $\{w^\star_j\}$. It is important to note that OLS is suboptimal in general for mixtures of linear regressions. Nonetheless, the transformer is performing much better than OLS, indicating the trained transformer implements a better predictor which is adapted to the mixtures of linear regressions setting.

**Extension to non-linear models:** In Appendix E, we also present result for the extension to non-linear mixtures of regressions. Qualitatively, we observe similar behavior where the trained transformer is competitive with oracle procedures which know the exact mixture model.

### 3.2 COMPARISON OF PERFORMANCE FOR FIXED TRAINING SET SIZE

Next, we investigate whether or not transformers learn mixtures of linear regressions in a sample efficient way. To do this, we depart slightly from the training methodology in (Garg et al., 2022).

---

[4]For interpretability of the figures, we omit the oracle algorithms above in the noiseless case ($\sigma = 0$) as the error is multiple orders of magnitude smaller than the data-driven procedures.

We first sample a fixed training set of size $n \in \{15000, 30000, 45000, 60000\}$. Then—with some hyperparameter tuning to avoid overfitting, as well as a modification to the curriculum training, described in Appendix C.1—we train the transformer as in that paper. We then compare the inference

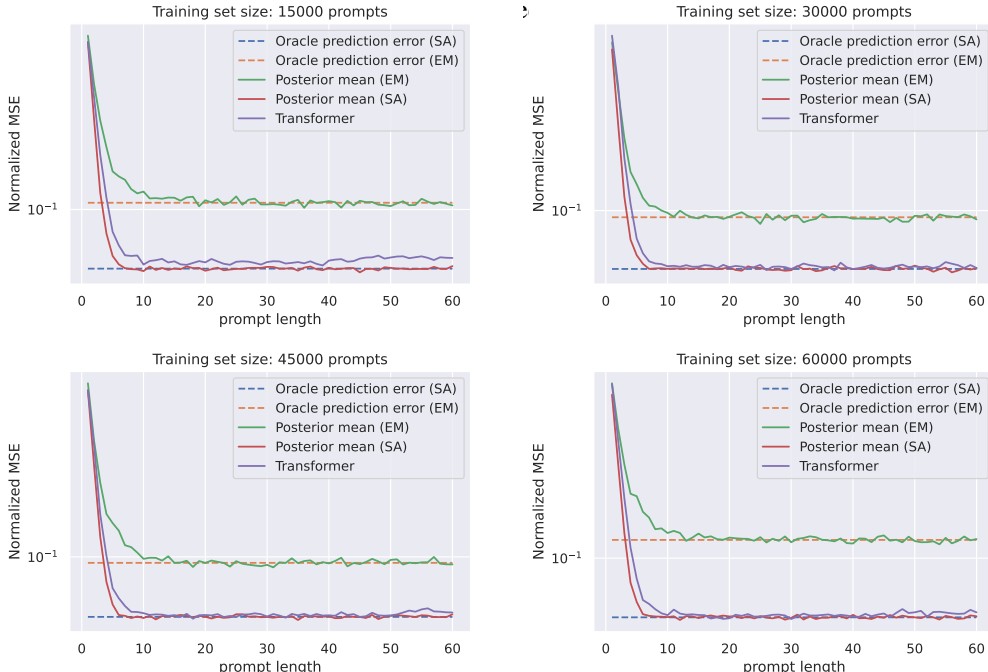

**Figure 3.** Comparison of EM, to subspace algorithm (SA) in (Jain et al., 2023), and Transformers.

The results of our simulation are shown below in Figure 3 We compared against two other procedures, which have the form of "plug-in" procedures:

- *Posterior mean, EM weights.* Here, we first estimate the component means $w_j^\star$ using batch expectation-maximiation (see Appendix C.2 and Algorithm 1 therein for details). Then, we form $\hat{\pi}$, the uniform distribution over the estimated weights, and then predict $y_{k+1}$ by $f_{\hat{\pi}}^\star(P)$.

- *Posterior mean, SA weights.* We follow the approach above, but estimate the weights by using the subspace algorithm (SA), which is Alg. 1 in (Jain et al., 2023).

Note that the 'oracle prediction error' quantities appearing in Figure 3 are essentially the best possible error achievable using the weights estimated by the set of weights $\widehat{\mathcal{W}}$ estimated by an algorithm. Before normalization by the dimension, it is the noise level plus

$$\frac{1}{m} \sum_{j=1}^{m} \min_{w \in \widehat{\mathcal{W}}} \|w_j^\star - w\|_2^2,$$

which is easily verified to be the prediction error with oracle knowledge of the nearest element in $\widehat{\mathcal{W}}$ to the true component mean $w_j^\star$, under our observational model (2). The main take-away from this simulation is that the transformer is able to get very close to the performance of the state-of-the-art model-specific algorithms, even when keeping the sample size the same.

### 3.3 What is the transformer actually learning?

In this section, we try to understand somewhat better, what algorithm the transformer is implementing at inference time. To do this, we define the squared distance, for two algorithms $f, g$ that map a prompt $P$ of length $k$ to a prediction $\hat{y}_{k+1}$ of $x_{k+1}$:

$$d_k^{\text{sq}}(f, g) := \mathbf{E}_P \left[ (f(P) - g(P))^2 \right], \quad \text{where } k \geqslant 1.$$

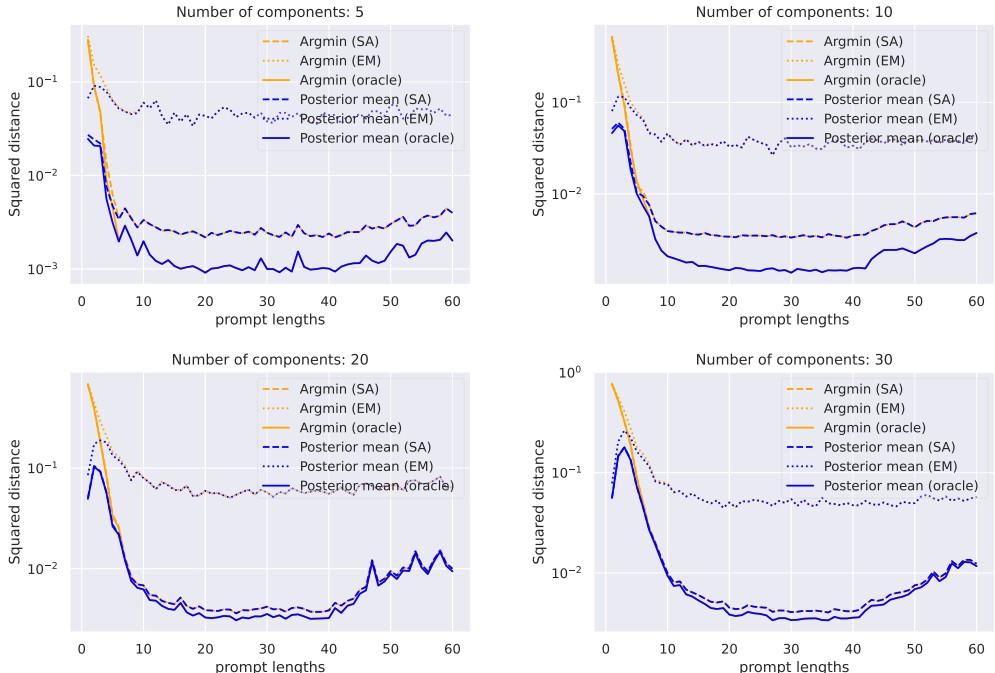

**Figure 4.** Comparing inference-time predictions from transformer versus posterior mean and argmin procedures with oracle or estimated weights. In these simulations, the noise level is set as $\sigma = 1.0$.

Figure 4 depicts $k$ versus $d_k^{\mathrm{sq}}(f, g)$, taking $f$ to be the transformer, and $g$ to be a candidate algorithm listed below, as $k$ varies between 1 and 60. The compared algorithms are:

- *Posterior mean, oracle weights.* Outputs $f_\pi^\star(P)$ on prompt $P$.
- *Posterior mean, SA weights.* Outputs $f_{\hat\pi}^\star(P)$, with $\hat\pi$ from the subspace algorithm (SA).
- *Posterior mean, EM weights.* Same as above, but $\hat\pi$ from expectation-maximization (EM).
- *Argmin, oracle weights.* Outputs $\hat{w}(P)^\intercal x_{k+1}$ where $\hat{w}(P)$ follows display (7).
- *Argmin, SA weights.* Outputs $\hat{w}(P)^\intercal x_{k+1}$ where $\hat{w}(P)$ follows display (7), with $w_j^\star$ replaced by SA-estimated weights.
- *Argmin, EM weights.* Same as above, but with EM-estimated weights.

As seen from Figure 4, in all of the simulated settings, the algorithm closest to the transformer at inference time is the posterior mean procedure, with the oracle weights. Impressively, this observation holds regardless of our choice of the number of mixture components.

### 3.4 Evaluation on covariate shift and label shift

In this section, we evaluate transformers on distribution shift settings. The experimental results are presented in Figures 5 and 6. The distribution shift settings are described below, where we studied one setting of covariate shift and two settings of label shift.

**Covariate scaling:** We evaluate the transformer on prompts of length $k$ where the covariates (including the query) are sampled as $x_i \sim \mathsf{N}(0, \kappa^2 I_d)$ for $i \in [k+1]$. This is a shift from the training distribution when $\kappa \neq 1$. Figure 5 shows the results when taking $\kappa \in \{0.33, 0.5, 1, 2, 3\}$. As we see from the figure, the transformer is able to handle, to some extent, small shifts, such as $\kappa \in \{0.33, 0.5, 2\}$, but not shifts much larger than this.

**Weight scaling:** We evaluate the transformer on prompts sampled from the mixture distribution

$$\pi_\alpha^{\mathrm{scale}} := \frac{1}{m} \sum_{i=1}^m \delta_{\alpha w_j^\star}, \quad \text{where } \alpha > 0.$$

So, the weights $w_j^\star$ are scaled up or down by the factor $\alpha$. Note that $\pi_1^{\mathrm{scale}} = \pi$, meaning that $\alpha = 1$ is no shift. The left panels of Figure 6 depict results for $\alpha \in \{0.33, 0.5, 1.0, 2, 3\}$.

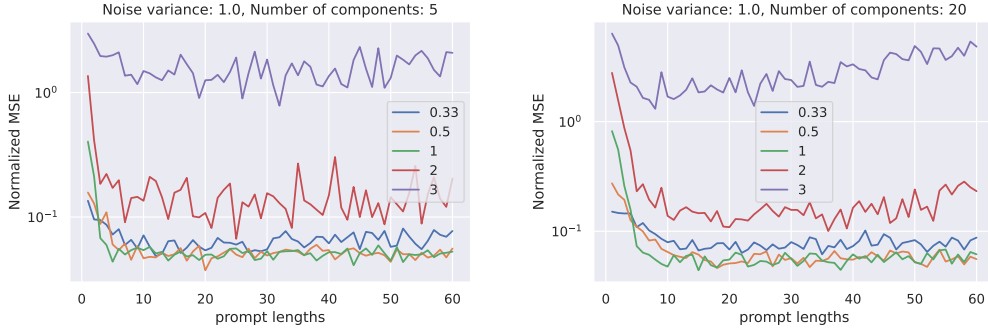

**Figure 5.** Evaluating transformer performance on covariate shifts.

**Weight shift:** We evaluate the transformer on weights sampled from the mixture distribution

$$\pi_\varepsilon^{\text{add}} \coloneqq \frac{1}{m} \sum_{i=1}^{m} \delta_{w_j(\varepsilon)}, \quad \text{where} \quad w_j(\varepsilon) \coloneqq w_j^\star + \frac{\varepsilon}{\sqrt{d}} \mathbf{1}_d.$$

Thus, $\pi_\varepsilon^{\text{add}}$ shifts each component by an additive perturbation of norm $\varepsilon$. The right panels of Figure 6 depict the results for $\varepsilon \in \{0, 0.25, 0.5, 0.75, 1.0\}$. Note that $\varepsilon = 0$ is no shift: $\pi_0^{\text{add}} = \pi$.

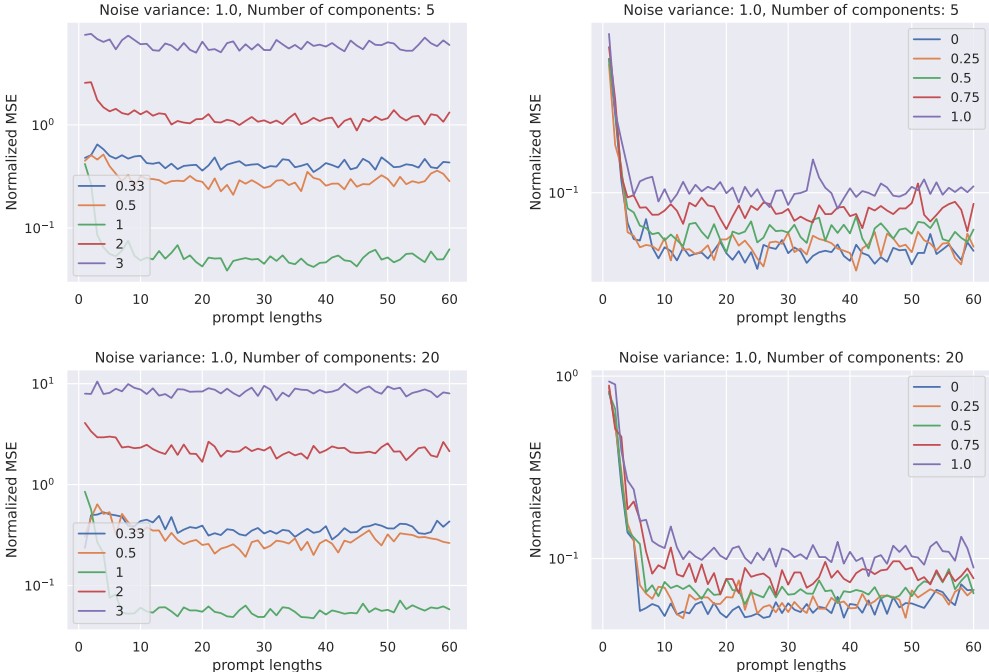

**Figure 6.** Evaluating transformers on weight scaling (left panels) and weight shift (right panels).

As seen from above, the transformer is fairly sensitive to weight scaling, as seen from the left panels in Figure 6. On the other hand, the transformer can handle small additive weight shifts, such as $\varepsilon = 0.25$, as depicted in the right panels in Figure 6. In Appendix F we also evaluate the minimum and maximum MSE components within the mixture distribution to show the sensitivity of the model to different inference-time mixture proportions.

**Comparison to posterior mean procedure:** In Appendix D, we replicate the figures above, with the change that in place of the transformer, we evaluate the performance of the posterior mean procedure, $f_\pi^\star$, defined in display (4). At a high-level, the posterior mean algorithm is less sensitive to covariate scaling, but exhibits similar behavior to the transformer on the two label shift settings.

## 4 DISCUSSION

In this work, we studied the behavior of transformers on mixtures of linear regressions, and showed they can learn these mixture models near-optimally, sample-efficiently, and robustly. The fact that general purpose predictors such as transformers can do well in this statistically-complex mixture setting should be practically useful, especially given its flexibility in the settings considered here.

Additionally, it would be interesting to study the in-context problem as was done in (Garg et al., 2022), but in the mixture setting. Here, the mixture distribution would be sampled from a distribution *over* mixture models for each prompt. In general, the decision-theoretic optimal method could be more complicated to compute, as implementing the posterior mean would require computing a high-dimensional integral. Nonetheless, is it possible to approximate the optimal method with a trained transformer? We view this as an intriguing direction for future work.

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

## A    RELATED WORK

The related work can be broadly divided into two categories: (i) theoretical works on learning mixture models and (ii) analyzing theoretically and empirically the learning abilities of transformers.

In the context of (i), there are numerous works that study the well known mixed linear regression problem with a batch size of 1 (Vempala & Wang, 2004; Yi et al., 2014; 2016; Chen et al., 2020; Li & Liang, 2018; Zhong et al., 2016). In general the problem is NP-Hard as shown in (Yi et al., 2016). Therefore most of the above works with the exception of (Li & Liang, 2018) makes the assumption that the covariates of all the mixture components are isotropic Gaussians. However, even with this strong assumption the time-complexity of all these algorithms are at least super-polynomical in $m$ rendering them impractical.

Kong et al. (2020) pioneered the study of the problem in batch setting where they were motivated by meta-learning multiple tasks. They showed that they can recover well separated mixture of linear models from batched data with polynomial dependence on $d, m$ and the inverse of the fraction of the smallest mixture component. However, this work still had the isotropic covariate assumption. Recent work (Jain et al., 2023) removed this assumption and further improved the sample complexity and the length of the medium size batches that is required for learning. We compare the training sample complexity of learning using transformers with that of the latter, as well as the popular EM method (Zhong et al., 2016), modified to work with batched data.

In the context of (ii), following the emergence of several hundred billion parameter large language models (LLM)'s like (Radford et al., 2019; Chowdhery et al., 2022), it has been observed that such models can learn from few examples supplied in a prompt during inference (Brown et al., 2020). This ability to learn in-context has been studied in simpler settings in many recent works (Garg et al., 2022; Von Oswald et al., 2023; Akyürek et al., 2023; Zhang et al., 2023). (Garg et al., 2022) showed empirically that transformers can learn to perform linear regression in context. (Akyürek et al., 2023) then showed that transformers can represent gradient decent for linear regression in context. A similar result was shown in (Von Oswald et al., 2023) but using linear self attention. Zhang et al. (2023) go one-step further by showing that gradient flow in linear self-attention based transformers can learn to do population gradient decent for linear regression. More general algorithm learning behavior has been demonstrated in (Li et al., 2023) and they also provide stability bounds for in-context learning.

Note that none of these prior works imply ability of transformers to learn mixture models from batch or non batch setting. Müller et al. (2021); Ahuja et al. (2023) look at in context learning from a Bayesian perspective. Müller et al. (2021) show that transformers fitted on the respective prior can emulate Gaussian processes. (Ahuja et al., 2023) has a section on learning multiple function classes in-context where they empirically study gaussian mixture models with two mixture components. However, they do not study the representation learning problem and training sample complexity is not investigated in depth.

The work in (Bai et al., 2023) is also extremely relevant to this work. They generalize the results in (Garg et al., 2022; Akyürek et al., 2023; Von Oswald et al., 2023) to show that transformers can represent a general version of in-context gradient descent which lets them show constructions for several in-context algorithms like linear, ridge regression and LASSO, gradient descent in 2-layer neural networks, in-context learning of GLMs. Perhaps most related is the part of the paper that deals with model selection. In particular the paper shows that tranformers can choose between two models classes either by a Post-ICL train-test split within context or a Pre-ICL distribution testing in-context. Note that these results do not generalize to our setting as (i) they do not yield our construction of the Bayes Opt in the mixture setting in Section 2 (ii) their experiments include learning with a mixture of different noise settings or distinguishing between a mixture of classification and regression, which is very different from having different $f_i^*$'s.

## B    PROOF OF THEOREM 1

In this section, we present the proof of Theorem 1. We begin, in Section B.1 by stating some preliminaries, such as the necessary operators we need to show that the transformer can implement. We then present the proof, assuming that these operators are transformer-representable in Section B.2.

Finally, the proof of the representation capacity of these operators by transformers is provided in Section B.3.

## B.1    OPERATORS THAT A TRANSFORMER CAN IMPLEMENT

We now list some operators, for a matrix $H \in \mathbf{R}^{p \times q}$ that output a matrix $H' \in \mathbf{R}^{p \times q}$. The following list includes all the operators we need.

- copy_down$(H; k, k', \ell, \mathcal{I})$:    For columns with index $i \in \mathcal{I}$, outputs $H'$ where $H'_{k':\ell',i} = H_{k:\ell,i}$, and the remaining entries are unchanged. Here, $\ell' = k' + (\ell - k)$ and $k' \geqslant k$, so that entries are copied "down" within columns $i \in \mathcal{I}$. Note, we assume $\ell \geqslant k$ and that $k' \leqslant q$ so that the operator is well-defined.

- copy_over$(H; k, k', \ell, \mathcal{I})$:    For columns with index $i \in \mathcal{I}$, outputs $H'$ with $H'_{k':\ell',i} = H_{k:\ell,i-1}$. The remaining entries stay the same. Here entries from column $i - 1$ are copied "over" to column $i$.

- mul$(H; k, k', k'', \ell, \mathcal{I})$:    For columns with index $i \in \mathcal{I}$, outputs $H'$ where
$$H'_{k''+t,i} = H_{k+t,i} H_{k'+t,i}, \quad \text{for } t \in \{0, \ldots, \ell - k\}.$$
for $t \in [k, \ell]$. The remaining entries stay the same.

- aff$(H; k, k', k'', \ell, \ell', \ell'', W, W', b, \mathcal{I})$:    For columns with index $i \in \mathcal{I}$, outputs $H'$ where
$$H'_{k'':\ell'',i} = W H_{k:\ell,i} + W' H_{k':\ell',i} + b.$$
Note that $\ell'' = k'' + \delta''$ where $W \in \mathbf{R}^{\delta'' \times \delta}$, $W' \in \mathbf{R}^{\delta'' \times \delta'}$ and $\ell = k + \delta$, $\ell' = k' + \delta'$. We assume $\delta, \delta', \delta'' \geqslant 0$. The remaining entries of $H$ are copied over to $H'$, unchanged.

- scaled_agg$(H; \alpha, k, \ell, k', i, \mathcal{I})$:    Outputs a matrix $H'$ with entries
$$H_{k'+t,i} = \alpha \sum_{j \in \mathcal{I}} H_{k+t,j} \quad \text{for } t \in \{0, 1, \ldots, \ell - k\}.$$
The set $\mathcal{I}$ is causal, so that $\mathcal{I} \subset [i - 1]$. The remaining entries of $H$ are copied over to $H'$, unchanged.

- soft$(H; k, \ell, k')$:    For the final column $q$, outputs a matrix $H'$ with entries
$$H'_{k'+t,q} = \frac{e^{H_{k+t,q}}}{\sum_{t'=0}^{\ell-k} e^{H_{k+t',q}}}, \quad \text{for } t \in \{0, 1, \ldots, \ell - k\}.$$
The remaining entries of $H$ are copied over to $H'$, unchanged.

The important property of the above list of operators is that can all be implemented in a single layer of a autoregressive transformer.

**Proposition 1.** *Each of the operators* copy_down, copy_over, mul, aff, scaled_agg, *and* soft, *can be implemented by a single layer of an autoregressive transformer.*

See Section B.3 for a proof of this claim.

## B.2    PROOF OF THEOREM 1

In this section, we present the proof of Theorem 1, assuming Proposition 1. We need to introduce a bit of notation:
$$\mathcal{I}_{\text{even}}(k) \coloneqq \{2j : j \in [k]\} \quad \text{and} \quad \mathcal{I}_{\text{odd}}(k) \coloneqq \{2j - 1 : j \in [k + 1]\}.$$
Additionally, we define $\mathcal{W}^\star \in \mathbf{R}^{m \times d}$ to have rows $w_i^\star \in \mathbf{R}^d$, which as we recall from (10), are the true mixture weights.

We begin by assuming that the input prompt $P$ is provided as $H^{(0)} \in \mathbf{R}^{(2d+4m+2) \times (2k+1)}$. This matrix is such that the only nonzero entries are $H^{(0)}_{1:d,2j-1} = x_j \in \mathbf{R}^d$ for each $j \in [k + 1]$. Additionally, $H^{(0)}_{1,2j} = y_j$ for each $j \in [k]$ Then, by leveraging the operators described above, we can see that $f^\star(P) = H^{(9)}_{2d+4m+2,2k+1}$, where the matrix $H^{(8)}$ is constructed by the following process:

- $H^{(1)} = \mathsf{copy\_down}(H^{(0)}; 1, d+1, d, \mathcal{I}_{\mathrm{odd}}(k))$

- $H^{(2)} = \mathsf{copy\_over}(H^{(1)}; d+1, d+1, 2d, \mathcal{I}_{\mathrm{even}}(k))$

- $H^{(3)} = \mathsf{copy\_down}(H^{(2)}; 1, 2d+1, 1, \mathcal{I}_{\mathrm{odd}}(k))$

- $H^{(4)} = \mathsf{aff}(H^{(3)}; d+1, 2d+1, 2d+2, 2d, 2d+1, 2d+m+1, \mathcal{W}^{\star}, \mathbf{1}_{d \times 1}, 0, \mathcal{I}_{\mathrm{even}}(k) \cup \{2k+1\})$

- $H^{(5)} = \mathsf{mul}(H^{(4)}; 2d+2, 2d+2, 2d+m+2, 2d+m+1, \mathcal{I}_{\mathrm{even}}(k))$

- $H^{(6)} = \mathsf{scaled\_agg}(H^{(5)}; -\frac{1}{2\sigma^2}, 2d+m+2, 2d+2m+1, 2d+m+2, 2k+1, \mathcal{I}_{\mathrm{even}}(k))$

- $H^{(7)} = \mathsf{soft}(H^{(6)}; 2d+m+2, 2d+2m+1, 2d+2m+2)$

- $H^{(8)} = \mathsf{mul}(H^{(7)}; 2d+2, 2d+2m+2, 2d+3m+2, 2d+m+1\{2k+1\})$

- $H^{(9)} = \mathsf{aff}(H^{(8)}; 2d+3m+2, d+1, 2d+4m+1, 2d+2m, d+1, 2d+4m+2, \mathbf{1}_m, 0, 0, \{2k+1\}))$

The process above is illustrated in Section B.2.1. By Proposition 1, each operation above is implementable by a layer of an autoregressive transformer. Therefore, this completes the proof.

**Discussion on the construction:** Note that in the operation $\mathsf{aff}$ used above, we do assume that the transformer stores the weights $\mathcal{W}^{\star}$. While at first glance this may seem like a strong assumption, we note that in our setting it is actually information-theoretically possible to learn the weights consistently as the number of prompts tends to infinity while keeping the prompt length size fixed.(For instance, see the paper Jain et al. (2023) and references in section 1.2 therein for more discussion.) We view it as an interesting line of future work to investigate to what extent the transformer stores approximate versions of the true mixture parameters $w_i^{\star}$ explicitly. The main goal of the above construction is simply to show that it is indeed possible to implement the optimal method viz-a-viz a transformer.

### B.2.1 ILLUSTRATION OF PROOF OF THEOREM 1

We illustrate the steps taken by the transformer to implement the softmax operation. To begin with, the matrix input to the transformer is modelled as below, in the case where $k = 2$. Below, $\tilde{y}_i = (y_1, 0, \ldots, 0) \in \mathbf{R}^d$. Throughout we only show the nonzero entries (*i.e.*, , missing rows and columns are always assumed 0). Then, our input is

$$H^{(0)} = \begin{bmatrix} x_1 & \tilde{y}_1 & x_2 & \tilde{y}_2 & x_3 \end{bmatrix}.$$

After the $\mathsf{copy\_down}$ operation, we have

$$H^{(1)} = \begin{bmatrix} x_1 & \tilde{y}_1 & x_2 & \tilde{y}_2 & x_3 \\ x_1 & 0 & x_2 & 0 & x_3 \end{bmatrix}.$$

After the $\mathsf{copy\_over}$ operation, we have

$$H^{(2)} = \begin{bmatrix} x_1 & \tilde{y}_1 & x_2 & \tilde{y}_2 & x_3 \\ x_1 & x_1 & x_2 & x_2 & x_3 \end{bmatrix}.$$

After another $\mathsf{copy\_down}$ operation, we have

$$H^{(3)} = \begin{bmatrix} x_1 & \tilde{y}_1 & x_2 & \tilde{y}_2 & x_3 \\ x_1 & x_1 & x_2 & x_2 & x_3 \\ 0 & y_1 & 0 & y_2 & 0 \end{bmatrix}.$$

After the $\mathsf{aff}$ operation, we have,

$$H^{(4)} = \begin{bmatrix} x_1 & \tilde{y}_1 & x_2 & \tilde{y}_2 & x_3 \\ x_1 & x_1 & x_2 & x_2 & x_3 \\ 0 & y_1 & 0 & y_2 & 0 \\ 0 & r_1 & 0 & r_2 & r_3 \end{bmatrix}.$$

Note that $r_i = \mathcal{W}^\star x_i - y_i \mathbf{1}$ for $i \neq k+1$ and otherwise $r_{k+1} = \mathcal{W}^\star x_{k+1}$. After the mul operation, we obtain

$$H^{(5)} = \begin{bmatrix} x_1 & \tilde{y}_1 & x_2 & \tilde{y}_2 & x_3 \\ x_1 & x_1 & x_2 & x_2 & x_3 \\ 0 & y_1 & 0 & y_2 & 0 \\ 0 & r_1 & 0 & r_2 & r_3 \\ 0 & r_1^2 & 0 & r_2^2 & 0 \end{bmatrix}$$

Above the square should be interpreted element wise on the vectors $r_i$. Then, after the scaled_agg operation, we obtain

$$H^{(6)} = \begin{bmatrix} x_1 & \tilde{y}_1 & x_2 & \tilde{y}_2 & x_3 \\ x_1 & x_1 & x_2 & x_2 & x_3 \\ 0 & y_1 & 0 & y_2 & 0 \\ 0 & r_1 & 0 & r_2 & r_3 \\ 0 & r_1^2 & 0 & r_2^2 & -\frac{1}{2\sigma^2}(r_1^2 + r_2^2) \end{bmatrix}.$$

Then, after the softmax operation, we obtain

$$H^{(7)} = \begin{bmatrix} x_1 & \tilde{y}_1 & x_2 & \tilde{y}_2 & x_3 \\ x_1 & x_1 & x_2 & x_2 & x_3 \\ 0 & y_1 & 0 & y_2 & 0 \\ 0 & r_1 & 0 & r_2 & r_3 \\ 0 & r_1^2 & 0 & r_2^2 & -\frac{1}{2\sigma^2}(r_1^2 + r_2^2) \\ 0 & 0 & 0 & 0 & p \end{bmatrix}.$$

Here, $p = \mathrm{softmax}(-\frac{1}{2\sigma^2}(r_1^2 + r_2^2))$. Finally, after yet another mul, we obtain

$$H^{(8)} = \begin{bmatrix} x_1 & \tilde{y}_1 & x_2 & \tilde{y}_2 & x_3 \\ x_1 & x_1 & x_2 & x_2 & x_3 \\ 0 & y_1 & 0 & y_2 & 0 \\ 0 & r_1 & 0 & r_2 & r_3 \\ 0 & r_1^2 & 0 & r_2^2 & -\frac{1}{2\sigma^2}(r_1^2 + r_2^2) \\ 0 & 0 & 0 & 0 & p \\ 0 & 0 & 0 & 0 & p \circ r_3 \end{bmatrix}.$$

Above, $\circ$ denotes elementwise multiplication. Finally, after an aff operation, we obtain

$$H^{(9)} = \begin{bmatrix} x_1 & \tilde{y}_1 & x_2 & \tilde{y}_2 & x_3 \\ x_1 & x_1 & x_2 & x_2 & x_3 \\ 0 & y_1 & 0 & y_2 & 0 \\ 0 & r_1 & 0 & r_2 & r_3 \\ 0 & r_1^2 & 0 & r_2^2 & -\frac{1}{2\sigma^2}(r_1^2 + r_2^2) \\ 0 & 0 & 0 & 0 & p \\ 0 & 0 & 0 & 0 & p \circ r_3 \\ 0 & 0 & 0 & 0 & f_\pi^\star(P) \end{bmatrix}.$$

Note that the bottom-right entry of $H^{(9)}$ contains the output $f_\pi^\star(P)$. In other words, the desired result if $H^{(9)}_{2d+4m+2,2k+1}$. Note that $H \in \mathbf{R}^{p \times q}$ where $p = 2d + 4m + 2$, and $q = 2k + 1$.

### B.3 Proof of Proposition 1

To begin with, we recall a few definitions, introduced in recent work (Akyürek et al., 2023).

**Definition 1** (RAW operator). The Read-Arithmetic-Write (RAW) operators are maps on matrices, $\mathbf{R}^{p \times q} \to \mathbf{R}^{p \times q}$,

$$\mathsf{RAW}_\bullet(H; \mathcal{I}, \mathcal{J}, \mathcal{K}, \Theta_\mathcal{I}, \Theta_\mathcal{J}, \Theta_\mathcal{K}, \pi) = H', \quad \text{where} \quad \bullet \in \{\otimes, \oplus\}.$$

Here $\pi$ is a causal set-valued map, with $\pi(i) \subset [i-1]$. The operators $\otimes, \oplus$ denote elementwise multiplication and addition, respectively. The entries of $H'$ are given by

$$H'_{\mathcal{K},i} := \Theta_\mathcal{K}\left(\Theta_\mathcal{J} H_{\mathcal{J},i} \bullet \left(\frac{\Theta_\mathcal{I}}{\max\{|\pi(i)|, 1\}} \sum_{i' \in \pi(i)} H_{\mathcal{I},i'}\right)\right), \quad \text{and} \tag{8a}$$

$$H'_{\mathcal{K}^c,i} = H_{\mathcal{K}^c,i}, \tag{8b}$$

for each $i \in [q]$. Note that above $\mathcal{K}^c = [p] \setminus \mathcal{K}$, and for some positive integer $r$, $\Theta_{\mathcal{I}} \in \mathbf{R}^{r \times |\mathcal{I}|}$, $\Theta_{\mathcal{J}} \in \mathbf{R}^{r \times |\mathcal{J}|}$, and $\Theta_{\mathcal{K}} \in \mathbf{R}^{|\mathcal{K}| \times r}$.

In Akyürek et al. (2023), they show that the RAW operator can be implemented in one autoregressive transformer layer.[5] They also argue that (with a slight change in parameterization) that the mul and aff operators are transformer-implementable. Therefore, we simply need to argue that the operators soft, copy_down, copy_over, and scaled_agg are all implementable by a transformer.

To begin with, note that, by inspection, we have, with $\delta = \ell - k$,

$$\mathsf{copy\_over}(H; k, k', \ell, \mathcal{I}) = \mathsf{RAW}_{\oplus}(H; [k, \ell], \varnothing, [k', k' + \delta], I_{\delta+1}, 0, I_{\delta+1}, \pi_{\mathcal{I}}) \tag{9a}$$

$$\mathsf{copy\_down}(H; k, k', \ell, \mathcal{I}) = \mathsf{RAW}_{\oplus}(H; [k, \ell], \varnothing, [k', k' + \delta], I_{\delta+1}, 0, I_{\delta+1}, \pi'_{\mathcal{I}}) \tag{9b}$$

$$\mathsf{scaled\_agg}(H; \alpha, k, \ell, k', i, \mathcal{I}) = \mathsf{RAW}_{\oplus}(H; [k, \ell], \varnothing, [k', k' + \delta], I_{\delta+1}, 0, \alpha I_{\delta+1}, \pi''_{\mathcal{I}, i}) \tag{9c}$$

Above, note that the intervals $[a, b]$ are just the integers between $a$ and $b$ (inclusive) and that we have defined

$$\pi_{\mathcal{I}}(i) = \begin{cases} \{i - 1\} & i \geq 2 \\ \varnothing & \text{otherwise} \end{cases}, \quad \pi'_{\mathcal{I}}(i) = \begin{cases} \{i\} & i \in \mathcal{I} \\ \varnothing & \text{otherwise} \end{cases}, \quad \text{and} \quad \pi''_{\mathcal{I}, i}(j) = \begin{cases} \mathcal{I} & j = i \\ \varnothing & \text{otherwise} \end{cases}.$$

Therefore, the displays (9) establish the following result.

**Lemma 1.** *The operators* copy_over, copy_down, *and* scaled_agg *are all implementable via the* RAW *operator in a single autoregressive transformer layer.*

Finally, in Section B.3.1 we demonstrate the following result.

**Lemma 2.** *The softmax operation is implementable by an autoregressive transformer.*

This completes the proof of Proposition 1.

### B.3.1  PROOF OF LEMMA 2

In order to implement the softmax operation, we need to introduce a few other operations:

- $\mathsf{div}(H; j, k, k', \ell, \mathcal{I})$:  For columns with index $i \in \mathcal{I}$, outputs $H'$ where $H'_{k'+t, i} = H_{k+t, i} / H_{j, i}$ for all $t \in \{0, \ldots, \ell - k\}$. The remaining entries of $H$ are copies as is into $H'$.
- $\mathsf{mov}(H; k, k', \ell, \mathcal{I})$:  For columns with index $i \in \mathcal{I}$, outputs $H'$ where $H'_{k'+t, i} = H_{k+t, i}$ for all $t \in \{0, \ldots, \ell - k\}$. The remaining entries of $H$ are copies as is into $H'$.
- $\mathsf{sigmoid}(H; k, k')$ :  In the final column $q$, outputs $H'$ with $H'_{k', q} = \frac{1}{1 + e^{-H_{k, q}}}$. The remaining entries of $H$ are copies as is into $H'$.

The operations div, mov are special cases of the same operations as introduced in the paper Akyürek et al. (2023). Thus, we only need to demonstrate that sigmoid is transformer-implementable. Assuming this for the moment, note that the softmax operation $\mathrm{softmax}$ is then implementable by the following sequence of operations. Let $H$ denote the input to the softmax layer, and let $s = H_{k:\ell, q}$. Using the affine operation (aff) together with the softmax operation (soft) we can compute the values $1/(1 + e^{s_i})$. Using the affine operation (aff) together with the div operation, we can invert these values to compute $e^{s_i}$. Finally, we can compute the sum of these values $S = \sum_i e^{s_i}$ with an affine operation (aff) and we can divide by this sum using another div operation. The result values are $e^{s_i} / \sum_j e^{s_j}$, which is the softmax of the vector $s$. A move operation (mov) then can move these values into the correct locations, $H_{k':(k'+\ell-k), q}$, as required.

Thus, to complete the proof, we need to show how to implement the sigmoid operation. For this, we can begin by using the affine operation to insert a value of 1 in the final column, and another affine operation to insert a $2 \times 2$ identity matrix in the first 2 columns of $H$. Then by selecting $W^{\mathrm{K}}$ and $W^{\mathrm{Q}}$ to select the identity matrix and to select $(H_{k, i}, 1)$, respectively, we can ensure that

---

[5]Note that in that paper, they more or less ignore the aggregate approximation error. Following this approach, we also ignore the lower order terms arising from the approximation of the underlying operations such as multiplication, as was done in Akyürek et al. (2023).

$W^{\mathrm{K}}H_{:i} = I_{2\times2}$ and $W^{\mathrm{Q}}h_i = (H_{k,i}, 1)$, and the corresponding softmax values in the self-attention layer are $s_i = (e^{H_{k+t,i}}/(1 + H_{k,i}), 1/(e^{H_{k,i}} + 1))$. By selecting $W^{\mathrm{V}}$ to select the identity matrix in $H_{:i}$, and $W^{\mathrm{C}}$ to select the first value of $s_i$ and place it in position $k' + t$, we can ensure that $a = 1/(e^{-H_{k,i}} + 1)e_{k'}$, where $e_j$ denotes the $j$th standard basis vector. This value is precisely the sigmoid, as needed. To place this value in the correct location, we simply set the feedforward network matrices $W^{\mathrm{in}}, W^{\mathrm{out}} = 0$. Then, to preserve the output, we need to delete the value $1$, and identity matrices placed into $H$ at the beginning; this can clearly be done by two affine operations.

## C    ADDITIONAL DETAILS ON TRAINING METHODOLOGY

Our training approach closely follows that of Garg et al. (2022) and Akyürek et al. (2023). After some hyperparameter optimization, we settled on the choice of hidden dimension of $256$, $8$ attention heads, and $12$ layers. We trained our transformers using Adam, with a constant step size of $0.1$. We used curriculum training, as in Garg et al. (2022), with the exception of Figure 3, where the sample size was fixed. Our curriculum phases were $2000$ steps each, with a batch size of $64$. The final stage of training had $250000$ steps with $64$ batches. All of our figures presented mean squared errors computed over batch sizes of $256$. The dimension of the original covariates was $d = 20$ throughout this paper.

### C.1    DETAILS ON FIXED SAMPLE-SIZE TRAINING

In this setting, we used hyperparameter tuning over the dropout parameter, $\rho \in \{0, 0.05, 0.1\}$, and found the following choices to be best, for Figure 3:

- for $n = 15000$, we took $\rho = 0.0$.

- for $n = 30000$, we took $\rho = 0.1$.

- for $n = 45000$, we took $\rho = 0.0$.

- for $n = 60000$, we took $\rho = 0.1$.

We also used curriculum training in this setup, but obtained the samples by subsampling the fixed dataset. This was done by first randomly sampling a batch from the full dataset, and then randomly dropping and shuffling the prefix of each prompt so as to obtain a prompt of the shorter, desired length. Otherwise, the entire procedure was the same as the other figures, as described above.

### C.2    BATCH EXPECTATION MAXIMIZATION (EM) ALGORITHM

Batch expectation-maximization is a variant of the standard expectation-maximization method (see, for instance, Section 14.5.1 in Bishop (2006)). For completeness, we describe the algorithm formally here. Note that $\phi$ denotes the standard univariate Gaussian pdf below. For notation, we also denote the prompts as

$$P^{(i)} \coloneqq \left(x_1^{(i)}, y_1^{(i)}, \ldots, x_k^{(i)}, y_k^{(i)}, x_{k+1}^{(i)}\right), \quad \text{for } i \in [n].$$

The algorithm is then stated below as Algorithm 1

---

**Algorithm 1** Batch expectation-maximization for a discrete mixture of linear regressions with Gaussian noise

---

**Require:** Length $k$ prompts $\{P^{(i)}\}_{i=1}^n$ noise variance $\sigma > 0$, number of components $m > 0$.

  *Initialize* $\pi^{(0)} \in [0,1]^m$, drawn uniformly on the probability simplex.

  *Initialize* $w_j^{(0)} \in \mathbf{R}^d$, drawn uniformly on the sphere of radius $\sqrt{d}$ for $j \in [m]$.

  *Initialize* $\gamma_{ij}^{(0)} = 0$ for all $i \in [n], j \in [m]$.

  **while** have not converged **do**

    update prompt-component assignment probabilities,

$$\gamma_{ij}^{(t+1)} = \frac{\pi_j^{(t)} \prod_{l=1}^k \phi\left(\frac{y_l^{(i)} - (x_l^{(i)})^\intercal w_j^{(t)}}{\sigma}\right)}{\sum_{j'=1}^m \pi_{j'}^{(t)} \prod_{l=1}^k \phi\left(\frac{y_l^{(i)} - (x_l^{(i)})^\intercal w_{j'}^{(t)}}{\sigma}\right)}, \quad \text{for all } i \in [n], j \in [m].$$

    update the marginal component probabilities by the formula

$$\pi_j^{(t+1)} = \frac{1}{n} \sum_{i=1}^n \gamma_{ij}^{(t+1)}, \quad \text{for all } j \in [m]$$

    update the parameter estimates by solving,

$$w_j^{(t+1)} = \arg\min_{w \in \mathbf{R}^d} \left\{ \sum_{i=1}^n \sum_{l=1}^k \gamma_{ij}^{(t+1)} \left(y_l^{(i)} - w^\intercal x_l^{(i)}\right)^2 \right\}, \quad \text{for all } j \in [m].$$

    update the iteration counter, $t \leftarrow t + 1$.

  **end while**

  **return** final set of component centers, $\{w_j^{(t)}\}_{j=1}^m$

---

In our implementation we stop (or declare the algorithm converged) if $t > t_{\max}$, or if

$$\max_j \min_{j'} \|w_j^{(t)} - w_{j'}^{(t-1)}\|_2 \leq \varepsilon.$$

In our experiments we took $t_{\max} = 20000$ and $\varepsilon = 0.001$.

## D  COMPARISON TO DISTRIBUTION SHIFT WITH THE POSTERIOR MEAN ESTIMATOR

In this section, we replicate the figures presented in Section 3.4, except we evaluate the distribution shift settings on the posterior mean procedure, $f_\pi^\star$ as defined in display (4).

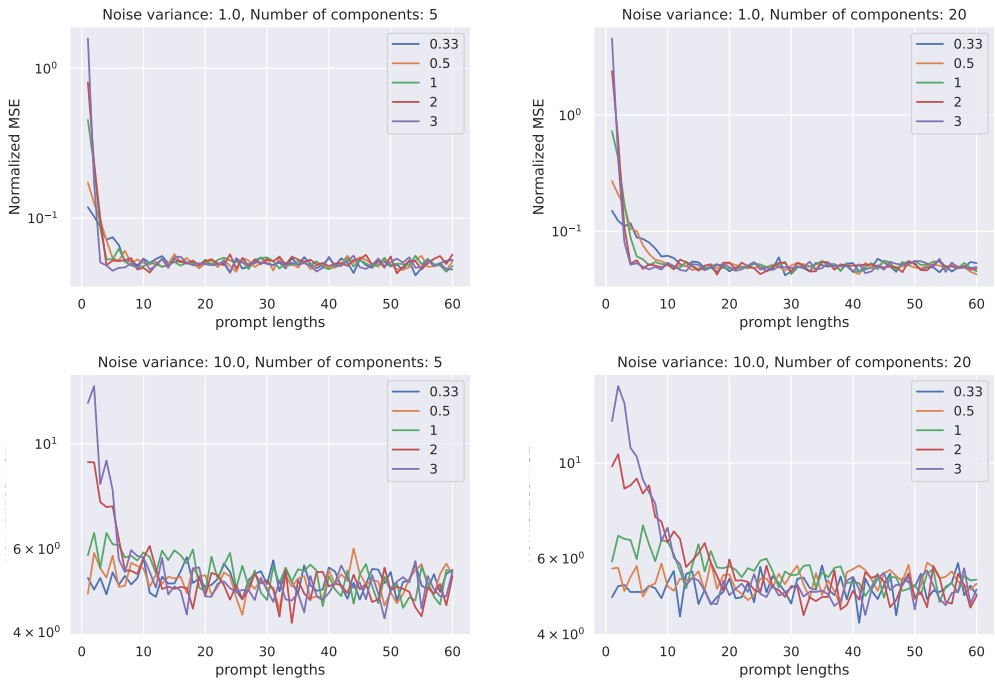

**Figure 7.** Posterior mean algorithm on covariate scaling distribution shift setting.

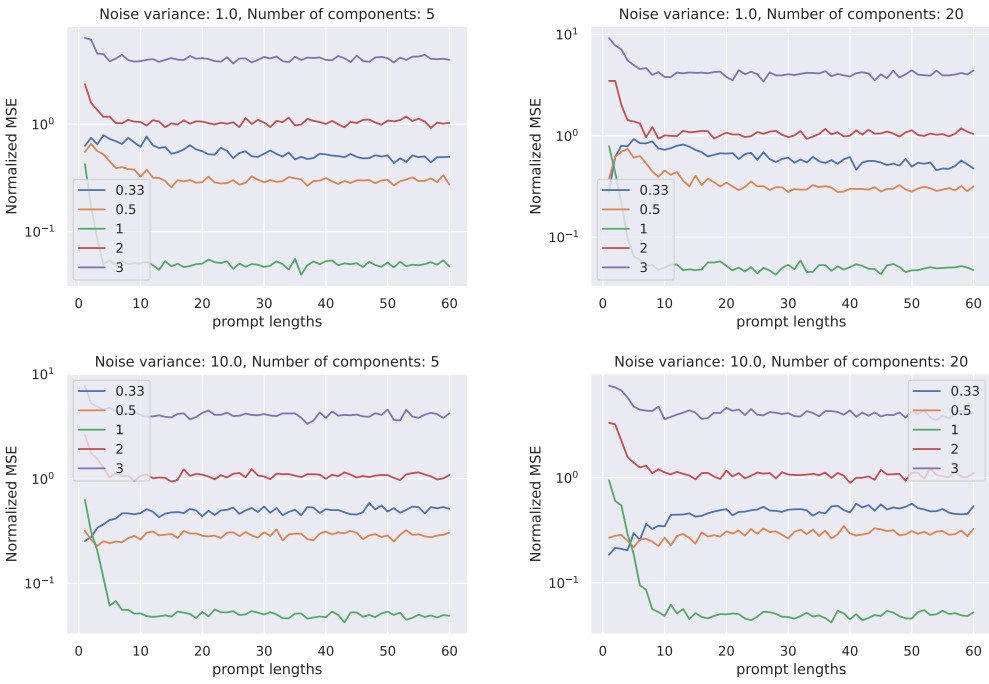

**Figure 8.** Posterior mean algorithm on weight scaling distribution shift setting.

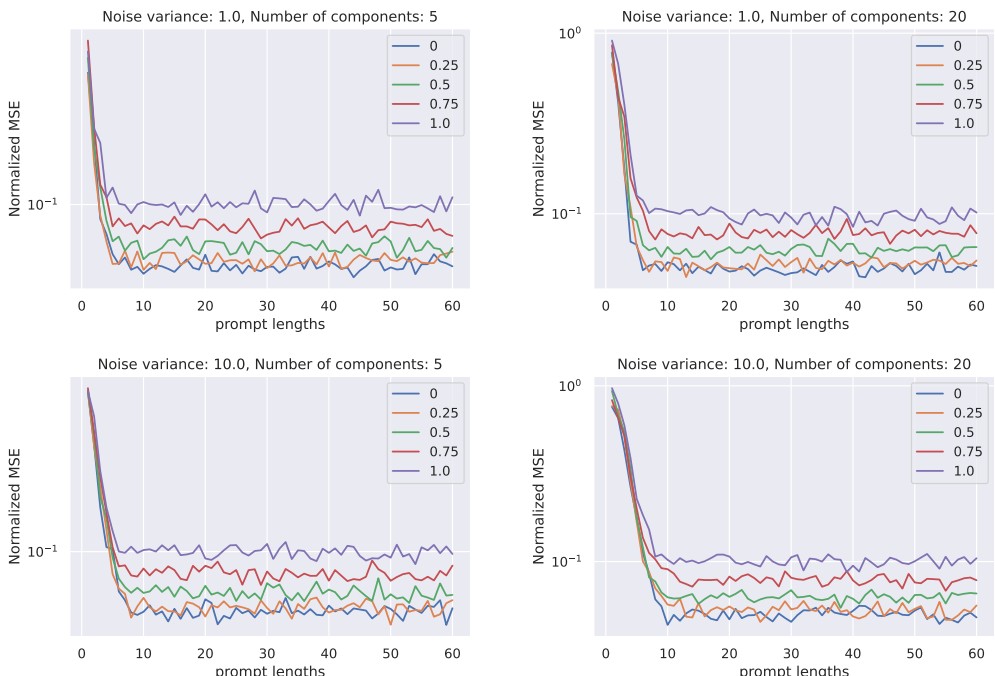

**Figure 9.** Posterior mean algorithm on weight additive shift setting.

## E   EXTENSIONS TO NON-LINEAR MIXTURES OF REGRESSIONS

In this section, we consider nonlinear extensions to the model described in Section 1.1.

We consider discrete mixtures of the form

$$\pi \coloneqq \frac{1}{m} \sum_{i=1}^{m} \delta_{f_i^\star}, \tag{10}$$

where $\{f_i^\star\}_{i=1}^m$ are functions lying a nonlinear function class $\mathcal{F}$, mapping $\mathbf{R}^d$ to $\mathbf{R}$. We consider *prompts* or *batches*, denoted $P = (x_1, y_1, \ldots, x_k, y_k, x_{k+1})$. Here, for noise level $\sigma \geqslant 0$, we have

$$f \sim \pi, \quad x_i \overset{\text{i.i.d.}}{\sim} \mathsf{N}(0, I_d), \quad \text{and} \quad y_i \mid x_i \sim \mathsf{N}(f(x_i), \sigma^2). \tag{11}$$

The goal is then to predict $y_{k+1}$, the label for the query $x_{k+1}$.

We train the transformer models here as described in the main text, with the exception that the data is generated according to the distribution above. We also compare, at inference time, to the following generalizations of the posterior mean algorithm, denoted $\hat{f}_{\mathrm{PMA}}$, and the argmin procedure, denoted $\hat{f}_{\mathrm{AM}}$. These are maps from the observed prompt $P$ to a function $f \in \mathcal{F}$, given by

$$\hat{f}_{\mathrm{PMA}}(P) \coloneqq \frac{\sum_{j=1}^{m} f_j^\star \exp\left(-\frac{1}{2\sigma^2} \sum_{i=1}^{k}(f_j^\star(x_i) - y_i)^2\right)}{\sum_{\ell=1}^{m} \exp\left(-\frac{1}{2\sigma^2} \sum_{i=1}^{k}(f_\ell^\star(x_i) - y_i)^2\right)}, \quad \text{and,}$$

$$\hat{f}_{\mathrm{AM}}(P) \coloneqq \arg\min_{f \in \{f_j^\star\}_{j=1}^m} \left\{ \sum_{i=1}^{k}(f(x_i) - y_i)^2 \right\}.$$

Thus, in order to specify the setting fully, we need only define our particular choice of function class, $\mathcal{F}$. We do this in each of the subsequent sections.

### E.1 MULTIVARIATE POLYNOMIALS

In this section, we consider degree-2 polynomials in $\mathbf{R}^d$ given by

$$\mathcal{F} = \Big\{ f(x) = \sum_{i,j=1}^{d} \alpha_{ij} x_i x_j : \alpha_{ij} \in \mathbf{R} \Big\}$$

To compute these efficiently, we select $\alpha_{ij} = w_i w_j$ where the weights $w_i \in \mathbf{R}$ are such that $(\sum_{i=1}^{d} w_i^2)^2 = d$, which ensure that $\mathbf{E}(y^2) = 3d + 1$. When plotting the mean squared error (MSE) in the figures below, we divide by $3d + 1$. The interpretation is that when the normalized MSE is equal to 1, we are doing no better than predicting 0 for each $x$; MSE significantly lower (on the normalized scale) indicates a substantial improvement.

In addition to the comparison with the posterior mean and argmin procedures, we also compare against polynomial regression, where $y$ is regressed on to the degree-2 monomials of the form $x_i x_j$, where $i, j \in [d]$. We additionally present results with a ridge penalty at the noise level.

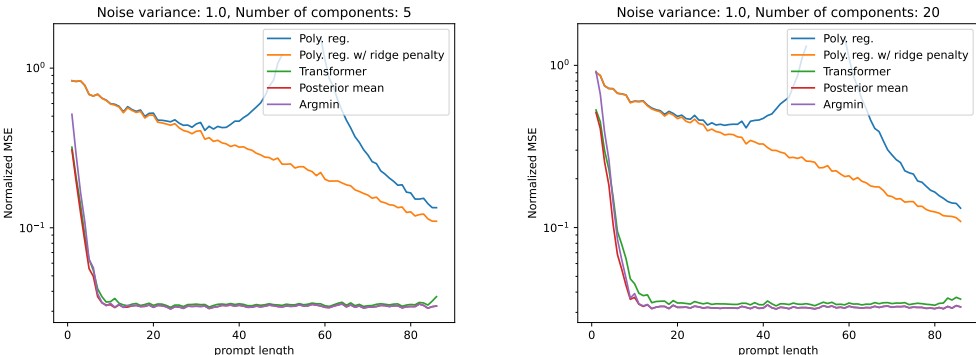

**Figure 10.** Transformer model trained on mixture of polynomial regressions data with 5 and 20 components, presented with added noise($\sigma = 1$).Comparison of trained transformer to posterior mean and argmin algorithms. We also compare against degree-2 polynomial regression with and without a ridge penalty at the noise level.

### E.2 2 LAYER FULLY-CONNECTED NEURAL NETWORKS

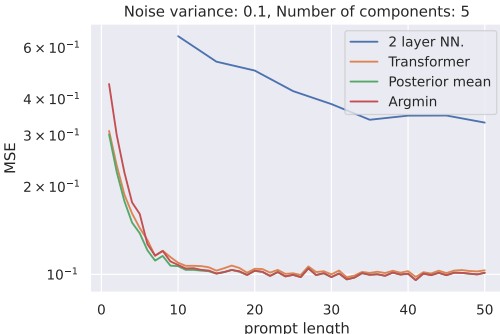

**Figure 11.** Transformer model trained on mixture of 2 layer NN regression data with 5 components, presented with added noise ($\sigma = 0.1$). Comparison of trained transformer to posterior mean and argmin algorithms. We also compare against 2 layer MLP regression.

We consider the case where $f_i^*$'s are 2 hidden layer MLP's similar. Note that (Garg et al., 2022) also had non-linear experiments with one hidden layer MLP's but not in the mixture setting. In particular our functions take the form,

$$f_i^*(x) = \sigma(W_i \sigma(U_i x + b_i^{(1)}) + b_i^{(2)}).\tag{13}$$

There are $m$ such fixed MLP's making the $m$ mixture components for generating the data. Each prompt is generated from one component after adding additive noise with a fixed variance. In Figure 11 we show the performance of our method against posterior mean and argmin procedure. We also include the result from regressing a single MLP (of the same size as data generating functions) per prompt – called 2 layer NN in the figure. The $f_i^{*}$'s used in this experiment have 10 neurons in both hidden layers. We set $m = 5$ and $d = 10$. The noise variance is set to $0.1$ which is roughly 10 percent of the signal for the networks chosen. We can clearly see that the trained transformer behaves very similar to the posterior mean algorithm which is optimal in this setting. Moreover, as expected the MLP regression does not work well since it trains one model per prompt.

## F    SENSITIVITY TO DIFFERENT MIXTURE PROPORTIONS

In this section, we extend our simulation results by considering what happens if the inference-time mixture propotions change. Consider a mixture distribution of the form

$$\pi = \sum_{i=1}^{m} p_i \delta_{w_i^{\star}},$$

where the weights $w_i^{\star}$ are the same as in the model (10). Note that the mean squared error of any algorithm under $\pi$ is given by the average

$$\text{MSE}(\pi) = \sum_{i=1}^{m} p_i \, \mathbf{E}_{w_i^{\star}} \left[ (f(P) - y_{k+1})^2 \right],$$

where the expectation under $w_i^{\star}$ indicates the distribution of prompt $P$ under the generative model (2) with the prompt weight fixed to be $w_i^{\star}$.

Evidently, by considering the maximum and minimum choice of mixture distribution $\pi$, we have

$$\min_{i \in [m]} \mathbf{E}_{w_i^{\star}} \left[ (f(P) - y_{k+1})^2 \right] \leqslant \text{MSE}(\pi) \leqslant \max_{i \in [m]} \mathbf{E}_{w_i^{\star}} \left[ (f(P) - y_{k+1})^2 \right].$$

This holds for any choice of mixture distribution $\pi$ where the weights are $w_i^{\star}$ but the mixture proportions $p_i$ are arbitrary.

Therefore, motivated by this inequality, we plot the maximum and minimum MSE for each algorithm we considered in the main text, for each choice of prompt length. The results are presented in Figure 12. The main takeaway is that the models are fairly insensitive to changing the mixture proportions, as indicated by the minimum and maximum of the MSE for each model being quite close to each other.

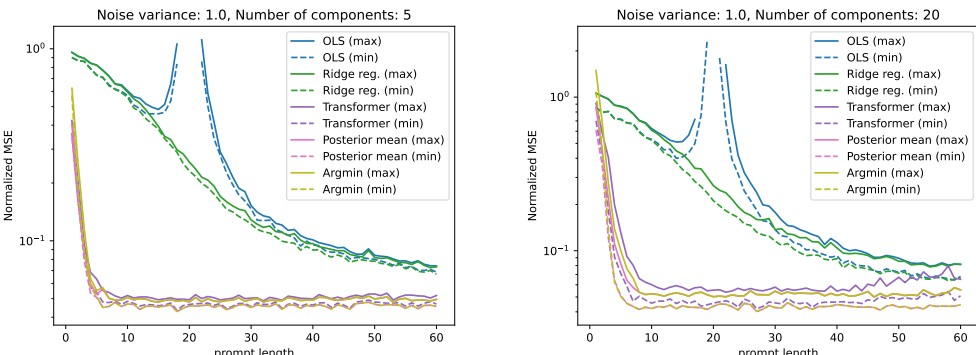

**Figure 12.** Transformer model trained on mixture of linear regressions data with 5 and 20 components. We also compare to ridge regression with the optimal penalty taken from either Corollary 3 in Dicker (2016) or Section 3.1.1 in Pathak et al. (2023). Each algorithm has two lines to indicate the component with the maximum and minimum MSE, for each prompt length.

## G  COMPARISON TO LINEAR REGRESSION METHODS AS $m$ GROWS LARGE

When the number of components $m$ grows large, the optimal method, when $\sigma > 0$, is a ridge regression estimator (asymptotically, as the prompt length grows large); this is justified by Corollary 3 in Dicker (2016) and the result in Section 3.1.1 of Pathak et al. (2023).

Thus, we investigate to what extent this is true in our experiments, with transformers. In Figure 13 we plot the mean squared errors of ridge regression divided by the mean squared error of our trained transformer. The training of the transformer follows the main text of the paper, and the ridge regression estimates were computed for each $k$ and $m$ on a batch of $2048$ samples. The qualitative takeaway is that as $m$ grows large, this ratio of risks is getting smaller. Since when $m$ is large the optimal estimator the optimal estimator should be closer to ridge regression, this can be interpreted as an indication that the same architecture of transformers used in our paper is able to be adapted to any structure of mixture distribution with essentially no modifications.

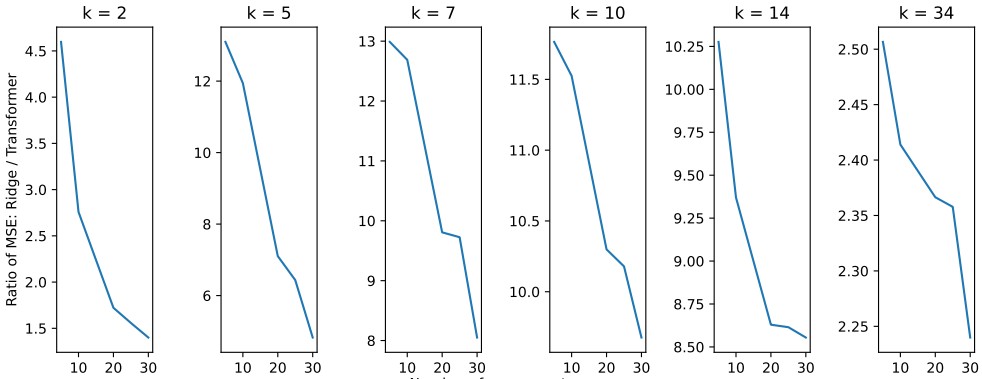

**Figure 13.** Comparison of the ratio of MSEs of ridge regression divided by the MSE of transformer for various choices of prompt length $k$ and number of mixture components $m$.

