# OpenReview forum: "Transformers can optimally learn regression mixture models"
_ICLR.cc/2024/Conference — ICLR 2024 poster_

### Official Review · Reviewer_sqwx · 2023-11-02

**Soundness:** 2 fair
**Presentation:** 3 good
**Contribution:** 3 good
**Rating:** 6
**Confidence:** 4

**Summary:**

This paper studies the ability of transformer models to perform mixture of linear regressions. Specifically, the authors train transformer models in the task of mixture of linear regressions, in which $k$ possible weight vectors are sampled with equal probability. They show that  there exists a decoder-based transformer that can implement the Bayes optimal for this task. Furthermore, they compare the performance of the transformer models with previously proposed algorithms.

**Strengths:**

This paper performs extensive experimental comparisons on the transformers' performance with other algorithms. They also extend the setting of previous work from linear regression to mixture of linear regressions.

**Weaknesses:**

It is unclear why this setting is of interest. I understand that transformers match the performance of proposed algorithms and that they could be used as an alternative for mixture of linear regressions. However, we should think of the cost training transformers and the time consumed, compared to performing any of the other algorithms.  The current models are used to perform language tasks and we are unaware on how optimization/numerical tasks could be merged with language tasks.

Furthermore, I think that more details should be provided for the experimental set-up.
Looking at the appendix of the paper, it is unclear to me how this proof is implemented. I think that the results of [1] require the design of specific encodings and entail some error, which was also not analyzed in [1], but the authors should at least show how it is controlled. I find in general the proof to be very high level and I had trouble verifying that it is correct.

[1]: Akyürek, Ekin, et al. "What learning algorithm is in-context learning? investigations with linear models." arXiv preprint arXiv:2211.15661 (2022).

**Questions:**

1. Did the authors perform experiments in which the weight vectors $w_i^*$ are not sampled with equal probability?
2. What models exactly the authors used? GPT2?
3. How many samples did they use to train the models and how many to test them? Which is also the sequence length for in-context learning? Is the sequence length during training the same as the one during inference?
4. How in the proof of lemma 2, the authors simply select $W_K H:,i = I_{2x2}$ since this matrix is required to have the second dimension equal to the sequence length.
5. Do the authors propose the use of transformers for this task ?
6. In the discussion section it is mentioned that "The fact that transformers... quite useful for practical problems". Could the authors mention some of those problems in which transformers would be useful ?

---

> ### Author Response · Authors · 2023-11-21
> **Thank you for your review; our comments and responses are below.**
>
> We thank you for your detailed comments and review.
>
> We would like to begin by pointing out that although transformers could be in the worst-case sense slower than other prediction methods, we found that in practice the transformers actually do perform significantly faster than the state of the art algorithms (such as the EM procedure and Subspace Algorithm used as comparisons in our work). This can potentially be explained by the fact that transformers can leverage GPU acceleration and parallelism that some of these other algorithms cannot.
>
> To answer your question (“do the authors propose the use of transformers for this task?”): indeed, we believe we are the first to propose transformers or mixtures of linear (or as in our updated revision, non-linear) regressions.  Moreover, we want to emphasize the novelty in our Section 3.2: we observe that not only can transformers adapt to unknown mixture models, they can also do so in a sample-efficient way. This is shown by keeping the sample size used to train the transformer as well as other state-of-the-art algorithms fixed and the same, and then showing that the transformer has essentially the same (or even better) performance at inference time. We also want to emphasize that in the non-linear setting, we are not aware of any algorithms which are able to guarantee good prediction of the labels, and surprisingly the transformer seems to be applicable to this setting with no additional changes to the underlying architecture.
>
> Moreover, we would like to point you to Appendix C which describes the answers to your questions about the models and training parameters. Our model is the same as that used in [2], and our training methodology only differs by the parameter choices described in Appendix C.1. If you have questions which are not addressed by that section, please let us know and we can easily update it in the revision. Finally, we have also provided our exact code along with the submission in case there is any concern regarding reproducibility.
>
> Finally, we thank you for your detailed reading of our original submission; we addressed your concern regarding the size of the identity matrix (it was a typo in the original submission), and added commentary regarding the hidden approximation errors as addressed in [1].
>
> [1] Akyürek, Ekin, et al. "​​ What learning algorithm is in-context learning? Investigations with linear models." The Eleventh International Conference on Learning Representations. 2022.
>
> [2]  Garg, Shivam, et al. "What can transformers learn in-context? a case study of simple function classes." Advances in Neural Information Processing Systems 35 (2022): 30583-30598.

---

> ### Author Response · Authors · 2023-11-21
> **One additional update, based on your questions.**
>
> Additionally, we want to thank you for the question regarding different weights for mixture components. In the updated revision, we have added a section (Appendix F) where we discuss the sensitivity of the models to different mixture proportions during inference. Since the MSE for any set of mixture proportions is upper (resp., lower) bounded by the maximum (resp., minimum) MSE for a component, we have plotted the performance of each algorithm for varying prompt lengths, on the worst and best mixture component. Any other mixture distribution with the same weights but different proportions must have MSE lying between these lines (see Figure 12). Additional discussion appears in Appendix F. The qualitative takeaway is that the models are generally not very sensitive to the choice of inference-time mixture proportions.

---

> > ### Comment · Reviewer_sqwx · 2023-11-23
> > **Further question**
> >
> > I would like to thank the authors for their response and extensive experiments. I would like to further ask about the implementation of the construction. It seems that the $w_i^*$ are known to the model to be able to represent the proposed estimator. I think that the real question is how the model extracts this optimal coefficients $w_i^*$, which they cannot be known a-priori. Do the authors have any intuition on this?
> >
> > I have updated my score to weak accept since the experiments are extensive, however I do not think that the provide theory is indicative of what the model actually learns.

---

> ### Author Response · Authors · 2023-11-23
> **Responding to your question.**
>
> Thank you for your very interesting question. Before responding to your main question, we also want to clarify where the weights are being stored in our construction. Our weights are stored by leveraging the affine operation in [1] (see top of page 14 in our draft). Since we leverage their construction, the mixture component parameters are stored in the feedforward part of the attention layer as in [1] (please see their equations (26) and (67) with Lemma 2 to verify this).
>
> Your main question, regarding “how the model extracts this optimal coefficients,” is certainly very interesting. However, we want to start by acknowledging that this is still open even in the case of simple linear regression (i.e., without any mixture distribution). Though better theoretical understanding of how transformers actually perform parameter estimation is important, we note that this is not a claimed direction of our work. More importantly, we feel strongly that it is quite far outside of the scope of this paper. Indeed, mixtures of (even linear) regressions is actually a challenging statistical setting that has led to numerous papers (for instance see section 1.2 in [3] for a list of over 20 recent papers in this area) and the fact that transformers can seamlessly solve this problem is—in our view—impressive. Moreover, our observations carry over to nonlinear settings as shown by Appendix E and as we show in section 3.2, this is also occurring in a sample efficient fashion.
>
> To further justify why we believe your question must be deferred to future work, we want to raise two important points. First, regarding the weights not being known “a-priori”: while it is certainly true that the weights are not known to the transformer a-priori, it is actually information theoretically possible to learn the weights consistently as the number of prompts tends to infinity while keeping the prompt length size fixed. (For instance, one can see this by applying Theorem 2 of the paper [2] in the case k = 1. This clearly also implies the existence of a consistent procedure when k > 1 as well, but one can also apply the main results in Section 2.4 of [3] when k > 1.) Therefore, although our construction requires storage of the exact weights, an interesting hypothesis to study in future work is that the transformer potentially stores an approximate version of the weights (up to possibly a linear transformation). To stress this point further, the papers we just cited also show that there is no theoretical barrier to this possibility.
>
> Secondly, your question is equally applicable to the construction in the paper [1]. Note that they also use their affine operation to guarantee that a transformer can implement 1-step of SGD (please see Appendix A, pg. 13 in [1]). There is no argument in [1] that such a mechanism is necessarily the one they observed occurring in practice, but rather they use the construction simplify to argue that transformers can carry out a sequence of computations. That is the same spirit in which we view our construction: it simply shows that there is a transformer which can carry out the sequence of operations needed to implement our optimal method. Of course, it is unclear that this is necessarily the one that is used in the final trained model (which is of course a very challenging question to address).
>
> We have also updated our paper by adding additional discussion on the construction at the end of the introduction in Section B.2. Finally, we want to thank you again for raising your question. We agree it is very important, but on the other hand, we also feel that our paper highlights numerous important findings for transformers as applied to the statistical challenging mixtures of linear and nonlinear regressions settings. These findings we believe should influence further theoretical progress and empirical work in this area.
>
> [1] Akyürek, Ekin, et al. " What learning algorithm is in-context learning? Investigations with linear models." The Eleventh International Conference on Learning Representations. 2022.
>
> [2] Kwon, Jeongyeol, and Constantine Caramanis. "EM converges for a mixture of many linear regressions." International Conference on Artificial Intelligence and Statistics. PMLR, 2020.
>
> [3] Jain, Ayush, et al. "Linear Regression using Heterogeneous Data Batches." arXiv preprint arXiv:2309.01973 (2023).

---

### Official Review · Reviewer_KcGJ · 2023-11-03

**Soundness:** 3 good
**Presentation:** 3 good
**Contribution:** 3 good
**Rating:** 8
**Confidence:** 3

**Summary:**

The authors explore whether transformer architectures can be trained to learn mixtures of linear models from batched data. They find that Transformers are surprisingly effective at this empirically, and they support their empirical results with a constructive proof that the optimal solution is representable in a transformer architecture.

**Strengths:**

I should note that this paper is out of area for me, and while I followed all the details, I may have missed some of the broader context.

I thought this a was well executed paper:
 - They give a nice constructive argument that shows that transformers can implement mixtures of linear regressions.
 - The experiments are very interesting in that they not only show that mixtures of linear regressions can be learned (this is perhaps not surprising given Garg et al's recent results showing this for linear regression), but that they are also competitive in terms of sample complexity with state of the art algorithms for this problem. I would have expected that you pay a larger cost for generality.

**Weaknesses:**

When I got to the end of the experimental section, I felt that there was a missed opportunity to look at whether Transformers allow one to easily go beyond the linear mixtures setting. While it is very interesting that Transformers are competitive with recent specialist algorithms for the linear mixture setting, I think the key advantage of a black-box method like a transformer is the ability to directly apply it to settings whether the linear mixture assumptions fail. Investigating how performance degrades (if at all) would have been interesting and would have potentially allowed you to show where Transformers outperform existing approaches.

This is brought up in the future work section of the discussion, but I think it would have been better to include it in this paper.

**Questions:**

I am most curious about whether you have experimented with any of the questions raised in your next steps. For example,

> in practice, the regression function within each component could potentially be nonlinear. To what extent do transformers perform well in these settings?

> In general, the decision-theoretic optimal method could be more complicated to compute, as implementing the posterior mean would require computing a high-dimensional integral. Nonetheless, is it possible to approximate the optimal method with a trained transformer?

I would be very happy to increase my score if you could show some experiments that show whether Transformers easily generalize beyond the linear Gaussian case (or not - a negative result could also be interesting if it is explained).

---

> ### Author Response · Authors · 2023-11-21
> **Thank you for your review; our comments and response are below.**
>
> Thank you very much for your detailed review.
>
> We wholeheartedly agree that the non-linear mixture setting is quite important in practice. Additionally this is indeed a key advantage to transformers: they can potentially flexibly extend to nonlinear data. To address your concern, we have taken your suggestion and presented in Appendix E extensive additional new experiments along these lines. There, we present new simulation results when the conditional mean of the label (given the covariate x) is either a multivariate polynomial or a function which itself is expressible via a multilayer perceptron. In both cases, our previous observations from linear mixture models do in fact carry over: the trained transformer is able to achieve nearly the Bayes-optimal risk (that is, the risk achieved by the oracle which knows the exact model used to generate the data). Please see Appendix E for further discussion and details.
>
> We believe that these experiments do support the claim that transformers can adapt to unknown (even nonlinear) mixture models, which should be of considerable interest to practitioners who work with data that is reasonably modeled as coming from a mixture distribution. In your original review, you had mentioned that you would “be very happy to increase [your] score” on the basis of such new experiments, so we hope that you will review the new experiments and let us know if you have additional concerns during the discussion period.

---

> > ### Comment · Reviewer_KcGJ · 2023-11-21
> >
> > Thanks for the updated experiments, I have increased my score.

---

### Official Review · Reviewer_1RLx · 2023-11-04

**Soundness:** 3 good
**Presentation:** 3 good
**Contribution:** 3 good
**Rating:** 6
**Confidence:** 3

**Summary:**

The paper studies the performance of transformers on in-context learning problems consisting of mixtures of linear regression models, where the prompt consists of input-output pairs coming from a linear model with one of $m$ different target weight vectors.
The authors show that the posterior mean in this problem may be implemented with a well-chosen transformer, and that empirically the predictions behave similarly to this algorithm, in particular outperforming OLS and EM approaches.

**Strengths:**

The paper makes an interesting contribution in the recently popular literature on using transformers for in-context learning regression models.

The posterior mean in eq.(4) is particularly interesting as a desired goal, as it requires a more complex transformer architecture than related papers, which combines both in-context algorithmic operations, and some "knowledge" from the data distribution, in the form of the $w_i^*$ vectors.

The experiments seem to give promising evidence that this target model may indeed resemble the one learned by transformers.

**Weaknesses:**

Two points would significantly strengthen the paper:

* while empirical results suggest the transformer might be related to the posterior mean, it would be good to have some interpretability results to assess whether this is true in practice, and if your construction in Theorem 1 is practically relevant: is there any evidence that the blocks shown in Figure 1 are actually being learned by the pre-trained transformer? Where are the $w_i^*$ being stored in the weights? Is it necessary to have at least 5 layers in practice, as in your construction? Is it sufficient?

* a more extensive empirical analysis would be useful, particularly on how the results vary when changing problem parameters. For instance, how does the performance change as $m$ varies? In particular, it seems difficult to find all the hidden directions $w_j^*$ once $m$ is too large -- does it start resembling OLS at some point? Does increasing the width, number of heads, number of layers change this?

* related work: is your setting covered by the general setting in [this paper](https://arxiv.org/abs/2306.04637)?

**Questions:**

see weaknesses

---

> ### Author Response · Authors · 2023-11-21
> **Thank you for your detailed review; our comments and response are below.**
>
> Thank you for your detailed review.
>
> We would like to begin by pointing out that the methodology we use in Section 3.2 to assess the similarity between the transformer and the posterior mean algorithm is essentially the same approach that is used in the paper [1] to compare multiple algorithms’ similarity. Additionally, similar to that paper, we construct a transformer which is potentially larger than necessary to implement the posterior mean algorithm (note that our mathematical construction requires more than 5 layers, as is shown in section B.2.1). Nonetheless, our models are much smaller than the theory suggests in the case of 20-component mixtures, which indicates that transformers can still perform very well even with smaller models than theory can potentially guarantee will succeed. This type of gap we leave open, as is done in [1], where they similarly have a gap between the size of the model implemented and the size of the model used in theory.
>
> Additionally, thank you for the pointer to the linked paper, we have added that paper to our extended related work, please see Appendix A. The paper is very insightful and definitely related to the  current work, however we believe our paper is substantially different from this paper. In particular, that paper does not consider mixtures of regressions of the form considered in this paper. (While they do consider 1 mixture-of-linear-regression setting, in that model they do not change the conditional mean, only the noise level. This makes their setup arguably easier than ours: for instance, ERM over all the data is consistent in their setting while it is not in ours). Moreover, that paper does not do any empirical comparison of sample complexity (such as we carry out in Section 3.2); sample efficient learning is essential in problems where one cannot collect additional data for free. Our results highlight that transformers are very general purpose, but more importantly and more strikingly, also very sample-efficient, since they achieve nearly the same error as state-of-the-art model-specific methods, with the same sample size.
>
> [1] Akyürek, Ekin, et al. "​​ What learning algorithm is in-context learning? Investigations with linear models." The Eleventh International Conference on Learning Representations. 2022.

---

> ### Author Response · Authors · 2023-11-22
> **One additional experiment added to address your question.**
>
> Thank you for your suggestion to investigate the relationship between the number of components, m, and the gap between the transformer and linear regression methods. First, we want to point out that it is more natural under our setup (where we have noise and sample weights from a distribution on a norm-constrained set) to consider the convergence of the transformer to ridge regression (with appropriately set penalty), as this is approximately equal to the Bayes estimator for this setup; see the papers [1, Corollary 3] and [2, Section 3.1.1] for additional details to justify this claim.
>
> Therefore, following your suggestion, we have now revised the paper to compare the ratio of the risk of transformer and ridge regression as a function of the number of components; please see Appendix G and Figure 13. The qualitative takeaway is: as the number of components grow, it is indeed true that the ratio of the risk of ridge regression to transformers is getting smaller, as your question predicted. Note also that in [3] they already show that transformers are close to OLS when sampling the weights from a Gaussian distribution. Note that in their setting, there is no noise, and so the optimal ridge estimator is in fact with penalty taken to be zero, i.e., OLS itself. Therefore, their observations are also consistent with our experiments: in the large m limit OLS and ridge regression are more competitive, but impressively, the transformer is able to adapt to this structure with out any modification to the model architecture.
>
> [1] Dicker, Lee H. “Ridge Regression and Asymptotic Minimax Estimation over Spheres of Growing Dimension.” Bernoulli, vol. 22, no. 1, Feb. 2016.
>
> [2] Reese Pathak, Martin J. Wainwright, Lin Xiao. “Noisy recovery from random linear observations: Sharp minimax rates under elliptical constraints.” arxiv, 2023
>
> [3] Garg, Shivam, et al. "What can transformers learn in-context? a case study of simple function classes." Advances in Neural Information Processing Systems 35 (2022): 30583-30598.

---

> > ### Comment · Reviewer_1RLx · 2023-11-23
> > **thank you**
> >
> > Thank you for your response. The new experiments are interesting, thanks for including them! I still think it would be good to interpret whether the trained model does resemble the proposed construction, and where the $w_i^*$ are being stored, but I understand that it's not an easy question. I am raising my score to 6.

---

> > > ### Author Response · Authors · 2023-11-23
> > > **Thank you for your response.**
> > >
> > > While we certainly agree understanding how exactly the transformer stores parameters is an important question, we do want to point out that this is not a claim in the current work. Indeed, we do not claim to show *how* the transformer learns the weights, only that (1) in experiments, it is getting very close to the optimal predictor and (2) it is possible for the transformer to actually implement the optimal predictor since it is within the model class (as shown by Theorem 1). While it is tempting to infer from (2) that the transformer actually does implement the optimal predictor in the manner we suggest, this is by no means our claim.
> > >
> > > Moreover, we want to point out that your question regarding if the "models does resemble the proposed construction" is equally applicable for the paper [1]. In that work, they only show that the transformer *can* implement the algorithms described therein, but they do not claim to show how that happens or even if it is occurring in the trained model. This is for a simple reason: it is very challenging to assess what exact way the transformer learns a given algorithm. Arguably, it is nonetheless extremely relevant for practical problems that the transformer is very close to the optimal predictor, and one should (in our opinion) view the construction as simply a proof of existence of the optimal predictor within the model class of all predictors implementable by an autoregressive (i.e., causal, decoder only) transformer.
> > >
> > > To further clarify this point, we have also added some additional discussion in the introduction of Section B.2 within the appendix. Thank you for your very interesting question again. While we agree it is very important, on the other hand, we also feel that our paper highlights numerous important findings for transformers as applied to the statistical challenging mixtures of linear and nonlinear regressions settings. These findings we believe should influence further theoretical progress and empirical work in this area.
> > >
> > > [1] Akyürek, Ekin, et al. " What learning algorithm is in-context learning? Investigations with linear models." The Eleventh International Conference on Learning Representations. 2022.

---

### Official Review · Reviewer_wAvt · 2023-11-06

**Soundness:** 3 good
**Presentation:** 3 good
**Contribution:** 2 fair
**Rating:** 6
**Confidence:** 3

**Summary:**

The authors argue that transformers provide a simple mechanism to efficiently and accurately learn mixtures of linear regression models.  They prove that the optimal solution is representable by such transformers and then experimentally verify that sample complexity/accuracy is on par with existing methods for this task.

**Strengths:**

It is a simply presented, well-articulated problem and solution.  The presentation is clear and more or less self-contained.

**Weaknesses:**

I'm not sure that the new work really address limitations in the existing literature.  The main motivation is that existing methods are potentially brittle and that their theoretical guarantees do not extend to the model misspecification setting.  It isn't clear that this work really demonstrates much of an improvement in that regard.  As a result, it isn't clear to me what this approach really offers (other than perhaps simplicity?).  It too does not come with any guarantees more generally -- or maybe I have misunderstood?

Also, it also feels like this result is a bit preliminary and could potentially encompass a wider range of mixture models and statistical settings.

**Questions:**

Questions are in the weakness section above.

Minor typos/suggestions:
- "gradient descent would naturally extends"
- "definitions in display (4)." -> "definitions in (4)."
- "better predictor to adapted to the mixtures of linear regressions setting"

---

> ### Author Response · Authors · 2023-11-21
> **Thank you for your review; our comments and response are below.**
>
> Thank you for your review and your constructive comments.
>
> We would like to begin by addressing your question regarding the limitations that this work addresses. Indeed, we would like to point out that in many settings, such as federated learning, one has to select the choice of the number of components in the mixture model by hand. For instance see papers [1, Section 2] and [2, Section 4.2]. This is quite tedious, and it is indeed of interest to have general purpose prediction methods which do not need to deal with the unknown parameter of the number of components in a mixture model.
>
> Regarding additional benefits of our work, we believe that transformers is potentially desirable due to its ability to adapt to the unknown mixture structure. In our original experiments, this was shown by employing the exact same transformer architecture on models with different numbers of components (5 or 20) and different noise levels (0 or 1). In our revision we also show transformers can extend to nonlinear models (see more discussion below). In all of these cases, we show that our trained transformer can achieve the same error as the posterior mean and argmin algorithms which are the decision-theoretically optimal methods and are very strong comparisons since they are oracle methods that have knowledge of the model. This, we believe, is quite striking: transformers have absolutely no model knowledge, yet nonetheless are performing as well as methods which have exact knowledge of the precise model.
>
> Moreover, we believe the sample complexity comparison done in Section 3.2 is very novel. Most other transformer papers do not consider the sample complexity of achieving a transformer which achieves low generalization error. Our experiment fixes the training sample size and shows that indeed transformers are nearly as sample-efficient as recently proposed, quite complicated methods which are model-specific. This is quite relevant for practice, where one may not have the luxury of seeking unlimited data and must cope with a finite, and perhaps only moderate-in-size sample. We view these results as quite surprising: transformers have absolutely no explicit knowledge of the model structure and yet nonetheless are achieving error on par with state-of-the-art methods, and with exactly the same sample complexity.
>
> Finally, we have further strengthened our results to address your concern about the limited scope of our original set of statistical settings. For instance, in appendix E, we have done extensive additional experiments, where we present new simulation results when the conditional mean of the label given the covariate x is either a multivariate polynomial or a function which itself is expressible via a multilayer perceptron. In both cases, our previous observations from linear mixture models do in fact carry over: the trained transformer is able to achieve nearly the Bayes-optimal risk (that is, the risk achieved by the oracle which knows the exact model used to generate the data). Please see Appendix E for further discussion and details. We believe that these experiments do support the claim that transformers can adapt to unknown (even nonlinear) mixture models, which should be of considerable interest to practitioners who work with data that is reasonably modeled as coming from a mixture distribution.
>
> Finally, we addressed the typos you pointed out. Thank you for your very close reading of our initial submission.
>
> [1] Sattler, Felix, et al. “Clustered Federated Learning: Model-Agnostic Distributed Multitask Optimization Under Privacy Constraints.” IEEE Transactions on Neural Networks and Learning Systems, vol. 32, no. 8, Aug. 2021, pp. 3710–22.
>
> [2] Yishay Mansour, Mehryar Mohri, Jae Ro, Ananda Theertha Suresh. “Three Approaches for Personalization with Applications to Federated Learning.” arxiv, 2020.

---

### Official Review · Reviewer_Pvi3 · 2023-11-08

**Soundness:** 3 good
**Presentation:** 3 good
**Contribution:** 3 good
**Rating:** 8
**Confidence:** 3

**Summary:**

The paper shows that transformer architectures can implement a mixture of linear regressions. Namely, it can implement the optimal solution that uses the true underline model parameters. The authors showcase their claim via a sequence of experiments.

**Strengths:**

* The idea of the paper is original and novel.
* The paper is written well. The illustration of the proof adds to the clarity of it and gives a good intuition.
* In general, the experiment section is good (although I think it misses some things; but more on that in the next section).
* Code was provided and the results seem to be reproducible.

**Weaknesses:**

* I am not convinced about the significance of this work. To clarify that, I would like the authors to address the following questions, how and when can one use the observation in the paper? In the introduction, federated learning is mentioned as a possible application. But, federated learning systems do not use linear models and if they do the number of components is known in advance (which is arguably the biggest advantage of the proposed viewpoint in the paper). I acknowledge though that this work can be a stepping stone towards a mixture of non-linear models, which potentially can have more impact.
* I do not have experience with mixture models. But, to me, it seems that a clear advantage of using mixture models is the access to the underline model and the mixture components. This is somewhat lost here. Having an approximation for the posterior mean only is nice, but I assume that in many cases one wants to evaluate each mixture separately in order to make a choice.
* Unless I didn't understand something, a potential issue for taking the viewpoint of this paper is that training transformers can be demanding in computation and data. I would expect that in most cases one would like to use these types of models in exactly the opposite cases, i.e., small training sets with limited computation.
* In section 3.1, under the noisy case, it is not surprising that the OLS model does not work as well since it doesn't model the noise. In my opinion, a more appropriate comparison would be an OLS model with a hyper-parameter for the noise variable which is chosen based on a validation set. Especially in light of the fact that a grid search was done for the dropout rate of the proposed approach. Conversely, and perhaps even more appropriate, is to compare to a Bayesian model and either optimize the noise via the marginal likelihood (or ELBO) or give it a full Bayesian treatment.
* To complement the question of "What is the transformer actually learning?" in section 3.3., I believe that some form of evaluation on out-of-distribution data be done. I suspect that the transformer learns an approximation for the posterior mean only in regions of in-distribution, but outside of it, it will behave in an arbitrary fashion. On the other hand, we know exactly how the posterior mean solution will behave in every region. Perhaps the authors can verify that using a similar experiment to the one in section 3.3 or on a simple 2D problem. If that is indeed the case, then how can you guarantee that indeed the solution found by the transformer matches that of the posterior mean?

**Questions:**

.

---

> ### Author Response · Authors · 2023-11-21
> **Thank you for your detailed review; our comments are below.**
>
> Thank you for the detailed review.
>
> We would like to begin by addressing your comments on federated learning. We want to point out that in federated learning applications, it is indeed not always the case that “the number of components is known in advance.” For instance, please see the papers [1, Section 3] and [2, Section 4.2]. In these works, the number of clusters is either learned or treated as a hyperparameter; it is not fixed ahead of time. In fact, as the latter paper says, it is often advisable to “find clusters for several values of [the number of clusters] and use the best one for each client separately using a hold-out set of samples.”
>
> Regarding motivation, we also wanted to point out that yet another motivation for this work can be found in pretraining LLMs, where one wishes to use diverse data (such as mixture data) with the hope that the transformer can zero-in on the relevant aspect (such as the correct mixture component). We study the simplest linear analogue of this. We also have revised the introduction to include this motivation as well as reference recent literature sharing in this motivation (see the blue text in the introduction of the revision).
>
> Regarding other predictors, we disagree that the reason the OLS model does “not work as well” is due to a lack of “model[ing] the noise.” In contrast, OLS is actually known to be decision-theoretically optimal in the scenario where the learner only has knowledge of the noise level (see for instance paper [3, Theorem 1] below). Instead, it is due to the norm constraint we employ on the linear regression model that improvements can be had (along with, of course the Bayesian structure of the generative process: a discrete mixture). Thus, we do agree with you that a better procedure would use the structure of the parameter (although, notably this is a somewhat unfair comparison, since the transformer does NOT have this knowledge).
>
> Nonetheless, following your suggestion we implemented the optimal ridge regression model under the assumption that the parameter is norm constrained, which consists of regularizing the parameter at the level of the noise level (see for instance the papers [4, Corollary 3] and [5, Section 3.1.1] below for justification of this claim). Moreover, as you seemed to suggest, this can also be seen as a MAP estimator under the Gaussian prior which concentrates on the scaled sphere in R^d. These results are now included in the revised version of Figure 2. Importantly, transformers still significantly outperform ridge regression.
> Note that the only change in the comparison qualitatively is that ridge regression is better than OLS when the prompt length is similar to the ambient dimension.
>
> Regarding other comparisons, please also note that throughout our paper we already compare the transformer to the optimal method which is the running posterior mean: see for instance our Figure 3. Note that the posterior mean is an oracle method since it knows the exact mixture distribution; surprisingly, the transformer is nonetheless able to achieve nearly the same performance as the posterior mean. This should be interpreted as: the transformer is doing essentially as best as possible for the settings described in our paper.
>
> Regarding out of distribution data, we would like to point out that figures 5 and 6, when compared to figures 8 and 9 in the appendix actually already comprises the comparison you asked for. We compare the performance of the transformer on various out of distribution settings and show that it behaves quite similarly to the posterior mean algorithm on these settings. We also have some discussion of this point in the paragraph immediately before the “Discussion” heading in the main text. To clarify one point, we also do agree that the “transformer learns an approximation for the posterior mean only in regions of in-distribution.” Nonetheless, the similarity to the performance of the posterior mean algorithm on covariate scaling experiments shows that it does gracefully degenerate in the same manner as the optimal in-distribution algorithm when evaluated in OOD settings. We would also like to point out that our methodology to justify the similarity of the transformer and the posterior mean is precisely the same as the paper [6].
>
> (continued in next comment)

---

> ### Author Response · Authors · 2023-11-21
> **Continuation of response to your original review.**
>
> (continuation from above)
>
> Regarding the practicality of our setting, we would also like to point out that in the revision, we have attempted to take yet a further step towards addressing more practical and nonlinear models. For instance, in Appendix E, we have done extensive additional experiments, where we present new simulation results when the conditional mean of the label (given the covariate x) is either a multivariate polynomial or a function which itself is expressible via a multilayer perceptron. In both cases, our previous observations from linear mixture models do in fact carry over: the trained transformer is able to achieve nearly the Bayes-optimal risk (that is, the risk achieved by the oracle which knows the exact model used to generate the data). Please see Appendix E for further discussion and details. We thank you for pointing out that nonlinear data “can have more impact” and hope the strong performance exhibited in our experiments can address this aspect of your previous concerns.
>
> [1] Sattler, Felix, et al. “Clustered Federated Learning: Model-Agnostic Distributed Multitask Optimization Under Privacy Constraints.” IEEE Transactions on Neural Networks and Learning Systems, vol. 32, no. 8, Aug. 2021, pp. 3710–22.
>
> [2] Yishay Mansour, Mehryar Mohri, Jae Ro, Ananda Theertha Suresh. “Three Approaches for Personalization with Applications to Federated Learning.” arxiv, 2020.
>
> [3] Mourtada, Jaouad. “Exact Minimax Risk for Linear Least Squares, and the Lower Tail of Sample Covariance Matrices.” The Annals of Statistics, vol. 50, no. 4, Aug. 2022.
>
> [4] Dicker, Lee H. “Ridge Regression and Asymptotic Minimax Estimation over Spheres of Growing Dimension.” Bernoulli, vol. 22, no. 1, Feb. 2016.
>
> [5] Reese Pathak, Martin J. Wainwright, Lin Xiao. “Noisy recovery from random linear observations: Sharp minimax rates under elliptical constraints.” arxiv, 2023
>
> [6] Akyürek, Ekin, et al. "​​ What learning algorithm is in-context learning? Investigations with linear models." The Eleventh International Conference on Learning Representations. 2022.

---

> > ### Comment · Reviewer_Pvi3 · 2023-11-22
> > **Response to authors**
> >
> > I would like to thank the authors for the answers and the additional experiments conducted following mine and other reviewers comments. In my opinion, the authors addressed most of the concerns raised adequately. Therefore, I decided to raise my score to by two levels to 8.

---

### Author Response · Authors · 2023-11-21
**Thank you for the detailed reviews; we have made many improvements based on your suggestions.**

We would like to thank all the reviewers for their detailed comments and suggestions.

Multiple reviewers requested extensions to our original experiments for non-linear mixtures of regression models. We would like to thank you for that very nice suggestion. We have done precisely that. In Appendix E of our revision, we present new, extensive simulations that show impressive behavior of transformers on nonlinear regressions. The qualitative takeaway is that transformers are indeed able to learn nonlinear mixtures of regressions, and in fact, they can do so without any tuning of the model architecture. We have taken our previous model architecture and applied it to polynomial regression problems as well as mixtures of MLP regression models (see Appendix E.1 and E.2); in all cases the transformer achieves the decision-theoretically optimal error on these problems. We believe this considerably strengthens our original submission, especially considering that for non-linear regression mixture models we are not aware of any algorithms in the literature that have been studied in our setting. Nonetheless, it was essentially effortless to extend transformers to this nonlinear setting due to the flexibility of transformers as general purpose prediction methods.

In addition to this update, we would like to emphasize two important points in our original submission. First, we would like to note that the sample-efficiency experiments we did in Section 3.2 are, to our knowledge, quite new and also, from our point of view, surprising. What it shows is that transformers are able to get nearly best-possible performance (better, or on par with that of state-of-the-art and model-specific methods for our setting) without any knowledge of the mixture distribution or even its structure. However, the most important point here is that this performance is achievable in a sample-efficient manner: Section 3.2 compares the transformers to other algorithms when they are both fed the same number of training data points. Thus, the takeaway is that our transformers not only learn an optimal predictor, but also do so sample-efficiently. This is not only surprising, but should also be of considerable interest to practitioners who are dealing with problems where they cannot just collect additional data for free.

Secondly, we want to address general questions regarding the motivation for our setting. Besides the fundamental interest in understanding how transformers behave in complex yet controlled statistical environments, we also believe that the model-agnostic nature of transformers is quite compelling given their excellent (and sample-efficient) performance and the fact that this extends across a variety of nonlinear and linear models. In federated learning, mixture models are used to leverage data from similar users in a network; one can motivate our setting by considering deploying transformer-based models on large scale networks without knowing in advance what the number of components is. In Bayesian learning, one may need to approximately compute a posterior mean over a prior on the parameter space which requires a high-dimensional integral; one can also motivate our setting by viewing the transformer as an flexible way to compute numerical approximations to these posterior mean estimators. Both, we believe, are compelling reasons to be interested in our setting. We have also pointed out in the revised introduction that such mixture models arise in pretraining LLMs.

All our revisions are highlighted in blue in the revised PDF. Thank you again for engaging comments and feedback. We look forward to continued discussion during the rebuttal period.

---

### Meta-Review · Area_Chair_r4Uu · 2023-12-09

**Metareview:**

This paper tries to answer the question whether transformers can exactly learn mixture models for regression. The authors provide a theoretical argument stating that there exist a transformer configuration able to learn certain arithmetic circuits encoding mixtures. Some empirical evidence on synthetic regression data.

The reviewers expressed mixed opinions, on the one hand raising concerns on the presentation and questioning the scope of the contribution, which seem limited, and missing details and baselines. On the other hand, they also appreciated the theoretical questions raised. During the rebuttal, the authors managed to flip two scores towards full acceptance by improving the presentation and answering the reviewers' concerns. In the discussion phase, reviewers still raised some concerns about the scope of the contribution.

The paper is accepted, but authors are asked to incorporate the reviewers' feedback and tone down their introduction and specifically rephrase "Is there a deep learning architecture that can be trained using standard gradient decent, yet learns mixture models from batched data and can leverage small batches from a source to make predictions for its appropriate subpopulation?" as the answer of that question is already yes, as a simple mixture model can already be represented as a differentiable computational graph.

**Justification For Why Not Higher Score:**

The scope of the paper is quite limited, and its impact is not clear at this stage.

**Justification For Why Not Lower Score:**

The paper could also be rejected. Some reviewers are indeed questioning its scope and the impact of the theoretical result.

---

### Decision · Program_Chairs · 2024-01-16

Accept (poster)